EMBO
Molecular Medicine

# A siRNA targets and inhibits a broad range of SARS-CoV-2 infections including Delta variant

Yi-Chung Chang[1], Chi-Fan Yang[2], Yi-Fen Chen[1], Chia-Chun Yang[2] (ID), Yuan-Lin Chou[2], Hung-Wen Chou[1], Tein-Yao Chang[3], Tai-Ling Chao[4], Shu-Chen Hsu[3], Si-Man Ieong[4], Ya-Min Tsai[4] (ID), Ping-Cheng Liu[3], Yuan-Fan Chin[3], Jun-Tung Fang[4], Han-Chieh Kao[4], Hsuan-Ying Lu[3], Jia-Yu Chang[3], Ren-Shiuan Weng[1], Qian-Wen Tu[1], Fang-Yu Chang[1], Kuo-Yen Huang[5], Tong-Young Lee[2], Sui-Yuan Chang[4,6,*] (ID) & Pan-Chyr Yang[7,8,9,**] (ID)

## Abstract

The emergence of severe acute respiratory syndrome coronavirus-2 (SARS-CoV-2) variants has altered the trajectory of the COVID-19 pandemic and raised some uncertainty on the long-term efficiency of vaccine strategy. The development of new therapeutics against a wide range of SARS-CoV-2 variants is imperative. We, here, have designed an inhalable siRNA, C6G25S, which covers 99.8% of current SARS-CoV-2 variants and is capable of inhibiting dominant strains, including Alpha, Delta, Gamma, and Epsilon, at picomolar ranges of $IC_{50}$ *in vitro*. Moreover, C6G25S could completely inhibit the production of infectious virions in lungs by prophylactic treatment, and decrease 96.2% of virions by cotreatment in K18-hACE2-transgenic mice, accompanied by a significant prevention of virus-associated extensive pulmonary alveolar damage, vascular thrombi, and immune cell infiltrations. Our data suggest that C6G25S provides an alternative and effective approach to combating the COVID-19 pandemic.

**Keywords** COVID-19; inhalation; K18-hACE2-transgenic mice; SARS-CoV-2; siRNA

**Subject Categories** Microbiology, Virology & Host Pathogen Interaction; Pharmacology & Drug Discovery

See also: **NAJ McMillan et al** (April 2022)

## Introduction

Severe acute respiratory syndrome coronavirus 2 (SARS-CoV-2) has infected over 200 million people and caused more than 4.5 million deaths worldwide as of September 13, 2021, according to the John Hopkins coronavirus resource center. Despite the availability of vaccines, breakthrough infections caused by the Delta variant appear to be driving a new wave of the pandemic (Dyer, 2021; Kupferschmidt & Wadman, 2021). The SARS-CoV-2 vaccines initially achieved great success in reducing viral infection and severe illness. The effectiveness of ChAdOx1 nCoV-19, BNT162b2, and mRNA-1273 vaccine reached 70.4, 95, and 94.1%, respectively (Baden *et al*, 2021; Knoll & Wonodi, 2021; Wang, 2021). Nevertheless, the emergence of new variants, especially the Beta and Delta variants, has raised great concerns, since reduced sensitivity of SARS-CoV-2 variants to therapeutic neutralizing antibodies, serum from convalescent patients, and vaccinated individuals have been reported (Planas *et al*, 2021). In addition, vaccine breakthrough cases have been described. A report from the UK indicates that the effectiveness of two doses of ChAdOx1 nCoV-19 vaccine against delta infection reduces to 67% while BNT162b2 reduces to 88% (Lopez Bernal *et al*, 2021). Another report from Qatar showed that two doses of BNT162b2 only present 51.9% of effectiveness while mRNA-1273 presents 73.1% (Tang *et al*, 2021). These reports implicate that viral mutations might influence vaccine effectiveness. Recently, a novel variant called Omicron, that contains twice the number of mutations in the spike protein compared with the Delta variant, has raised further concerns (Gao *et al*, 2021). Although the real-world effectiveness of vaccines against Omicron infection has not yet been released, reduced neutralizing activity of the Omicron

1  Oneness Biotech Company Limited, Taipei, Taiwan
2  Microbio (Shanghai) Biotech Company, Shanghai, China
3  Institute of Preventive Medicine, National Defense Medical Center, Taipei, Taiwan
4  Department of Clinical Laboratory Sciences and Medical Biotechnology, National Taiwan University College of Medicine, Taipei, Taiwan
5  Institute of Microbiology and Immunology, National Defense Medical Center, Taipei, Taiwan
6  Department of Laboratory Medicine, National Taiwan University Hospital and National Taiwan University College of Medicine, Taipei, Taiwan
7  Department of Internal Medicine, National Taiwan University Hospital and National Taiwan University College of Medicine, Taipei, Taiwan
8  Genomics Research Center, Academia Sinica, Taipei, Taiwan
9  Institute of Biomedical Sciences, Academia Sinica, Taipei, Taiwan
   *Corresponding author. Tel: +886 2 23123456 ext. 66908; Fax: +886 2 23711574; E-mail: sychang@ntu.edu.tw
   **Corresponding author. Tel: +886 2 23562905; Fax: +886 2 23582867; E-mail: pcyang@ntu.edu.tw

variant against mRNA vaccine-induced antibody responses has been reported (Edara *et al*, 2021), and a third dose of vaccination is currently suggested to provide robust neutralizing antibody responses against the Omicron variant. The emergence of Omicron variant indicates how fast the SARS-CoV-2 evolves, and its potential impact on the current protein-based intervention (i.e., vaccines, antibodies, or convalescent plasma) that primarily targets the highly mutated spike protein (van Dorp *et al*, 2020) cannot be neglected.

One therapeutic with great potential is short-interfering (si)RNA, which is an artificially synthesized double-stranded RNA of 19–23 nucleotides (Zamore *et al*, 2000). Upon entering the cytosol, siRNA interacts with several proteins to form an RNA-induced silencing complex (RISC) and subsequently knocks down the expression of target genes based on sequence complementarity. Bitko *et al* (2005) reported that intranasal instillation (IN) of unmodified naked siRNA is capable of inhibiting respiratory viral infection in mice without the help of any carrier or transfection reagent, IN and aerosol inhalation (AI) of naked siRNA have been widely utilized for delivering siRNA into pulmonary cells (Zafra *et al*, 2014; Kandil & Merkel, 2019). One successful application of an unmodified naked siRNA is ALN-RSV01 administered by nasal spray or aerosol inhalation to treat respiratory syncytial virus (RSV) infection. The phase II clinical trial showed that pretreatment with ALN-RSV01 significantly reduces the prevalence of RSV infection (DeVincenzo *et al*, 2010) and posttreatment with ALN-RSV01 reduces the risk of bronchiolitis obliterans syndrome in RSV-infected lung transplant patients (Gottlieb *et al*, 2016). These findings suggest the possibility of using respiratory-delivered, unmodified naked siRNA against SARS-CoV-2 infection. By targeting a highly conserved region of SARS-CoV-2, siRNA is capable of inhibiting a wide spectrum of viral variants and could thus be a one-for-all therapy for the rapidly evolving SARS-CoV-2.

However, unmodified naked siRNA is relatively vulnerable to nuclease degradation and could induce an innate inflammatory response through the activation of Toll-like receptors (TLR). The 2′-*O*-methyl and 2′-F modification of siRNA has been shown to reduce immune stimulation, including TLR-dependent and TLR-independent immune responses (Robbins *et al*, 2007; Meng & Lu, 2017). Here, a fully modified siRNA, C6G25S, was developed for safe, effective, and feasible therapeutics. We report the first study using fully modified siRNA with nasal drops or aerosol administration and, for the first time, effective use of siRNA therapy against the Delta variant strain *in vivo*.

# Result

## Selection and screening of a highly specific and potent siRNA against SARS-CoV-2

Coronavirus has the largest genome of all known RNA viruses, ranging from 26 to 32 kb (Woo *et al*, 2010). To identify a highly potent and specific siRNA sequence against SARS-CoV-2 variants, a systematic and comprehensive selection strategy was applied. As shown in Fig 1A, the filtering process began with a segmentation of the virus genome into 29,771 hit sequences of 19-nucleotide stretches. Next, 674 siRNA candidates with over 99.8% coverage rate among 29,871 SARS-CoV-2 genomes and their corresponding targeting regions

with low propensity for secondary structure were selected (Lan *et al*, 2020; Rangan *et al*, 2020). Furthermore, 374 siRNA candidates targeting to different regions of SARS-CoV-2 genomes were selected, including those in leader, papain-like protease, 3C-like protease, RNA-dependent RNA polymerase (*RdRp*), helicase, spike, and the envelope-coding regions (Kim *et al*, 2020). After removing those with high potential off-target effects on human transcriptome and targeting genes essential for cell viability, the top 11 siRNA with the lowest predicted off-target effects and highest predicted efficacy were selected and the detailed sequences with key comparison information are shown in Table 1. The effectiveness of selected siRNA to protect Vero E6 cells against SARS-CoV-2 infection was verified. *In vitro* screening in Vero E6 cells showed that C6, C7, C8, and C10 were capable of inhibiting up to 99% of both viral envelope gene expression and plaque-forming virion production at a concentration of 10 nM (Fig 1B and C).

The target sites of C6, C7, C8, and C10 on viral genome are presented in Table 1. C6, C8, and C10 were then fully modified into C6G25S, C8G25S, and C10G31A by 2′-*O*-methyl, 2′-fluoro, and phosphorothioate (PS) substitution for nuclease protection as depicted in Fig 1D. C7 was excluded from further analysis for sharing the same targeting gene with C6 and having more predicted off-target effects. Half-maximal inhibitory concentrations (IC50) for C6G25S, C8G25S, and C10G31A, determined by plaque reduction assay, were 0.07, 0.24, and 0.12 nM, respectively (Fig 1D). C6G25S was selected for subsequent *in vitro* and *in vivo* experiments for its lowest $IC_{50}$ value and numbers of off-target genes predicted in silico (Table 1). To evaluate the potential siRNA off-target effect, we performed two independent biological replicates of whole transcriptome sequencing from BEAS-2B cells transfected with the unmodified C6 and fully modified C6G25S, respectively. Whole transcriptome analysis showed that the modification of C6 could significantly reduce the total number of off-target genes in BEAS-2B cells from 51 (C6) to 21 (C6G25S) for two-fold expression differences (Appendix Fig S1). The 21 genes identified with two-fold expression differences after C6G25S treatment are listed in Table 2. Top 10 genes were confirmed by RT-qPCR. Moreover, C6G25S and unmodified C6 were found to have a similar $IC_{50}$ for inhibiting the viral RNA amplification (0.17 and 0.18 nM, respectively; Fig 1E). The expression of *RdRp*, the direct target for C6G25S, was also analyzed and revealed an $IC_{50}$ of 0.13 nM (Fig 1F). These data suggest that C6G25S is an siRNA with potent antiviral activity and a safety profile potentially beneficial for future drug development.

## C6G25S inhibits multiple strains of SARS-CoV-2 *in vitro*

According to the World Health Organization's website as of August 17, 2021, Alpha (B.1.1.7), Beta (B.1.351), Gamma (P.1), and Delta (B.1.617.2) are recognized as SARS-CoV-2 variants of concern (VOC), and Eta (B.1.525), Iota (B.1.526), Kappa (B.1.617.1), and Lambda (C.37) are recognized as SARS-CoV-2 variants of interest (VOI). C6 was designed to target a highly conserved region with no mutations from SARS-CoV-1 to SARS-CoV-2. The upper part of Fig 2A presents the location of C6, and the genetic map of the VOC, VOI, and other strains listed as indicated above. The lower part of Fig 2A shows the sequence alignment of C6 and *RdRp*, which is located in the 5-prime region of *ORF1b*. As observed in Fig 2B,

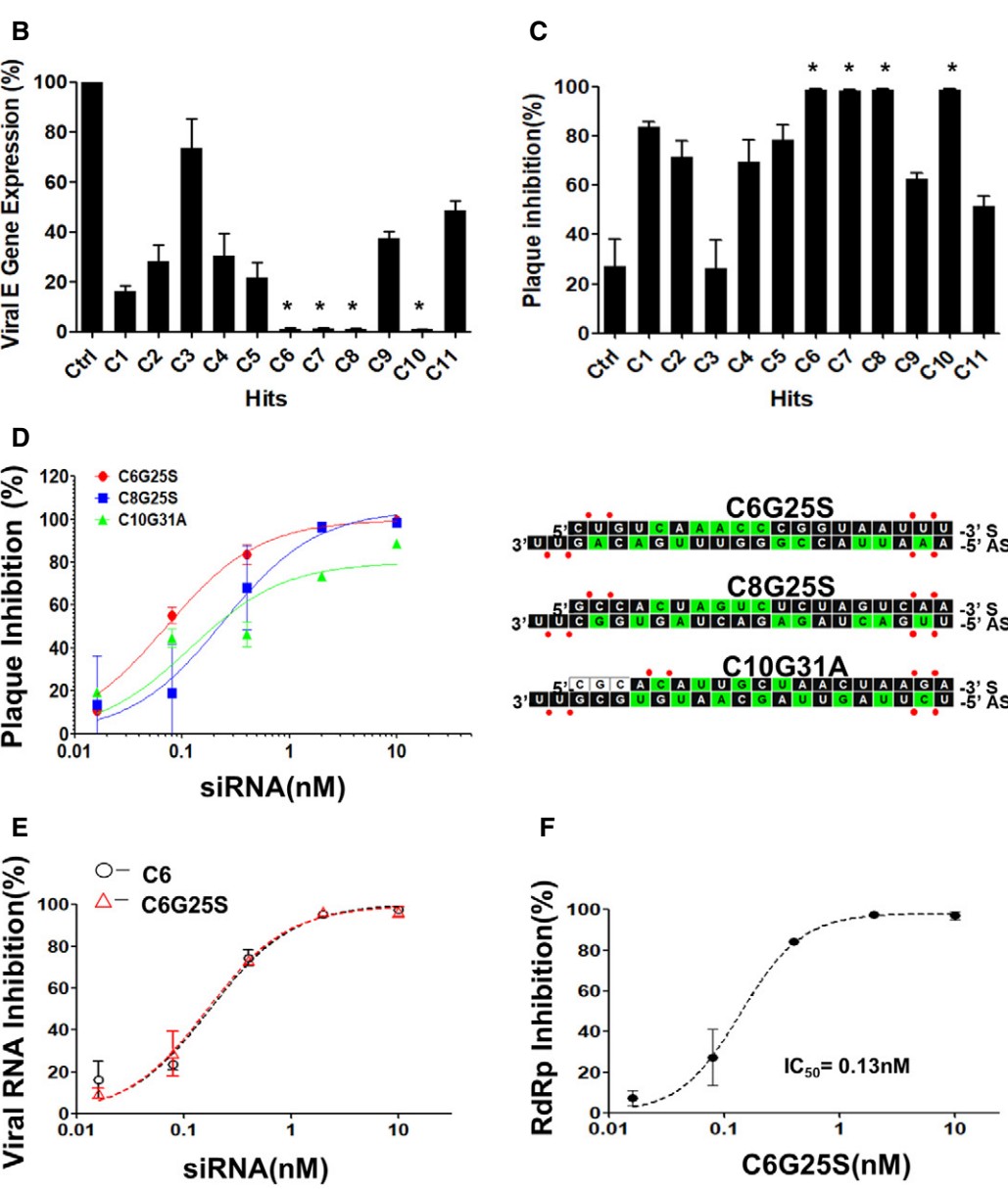

**Figure 1.**

**Figure 1. Selection of a highly potent siRNA against SARS-CoV-2.**

A Flowchart for the selection strategy. The selection criteria and sequence numbers remaining at the end of each stage were individually indicated.

B Vero E6 cells were transfected with 10 nM of siRNA before infection by SARS-CoV-2 at a multiplicity of infection (MOI) of 0.1 after 24 h. The numbers of viral RNA copies were quantitated with RT-qPCR. The negative control siRNA served as the control and is abbreviated as "Ctrl." C1–C11 represent the final candidate sequences after selection. The siRNAs capable of inhibiting up to 99% of viral envelope gene expression with $P$-values < 0.005 compared with Ctrl siRNA are marked with *. $P$-value by Student's $t$-test.

C Vero E6 cells were transfected with 10 nM of siRNA before infection by SARS-CoV-2 at a MOI of 0.1 after 24 h. The number of infectious virions were quantitated with plaque-forming assay. The siRNAs capable of inhibiting up to 99% of plaque-forming virions production with $P$-values < 0.0001 compared with Ctrl siRNA are marked with *. $P$-value by Student's $t$-test.

D $IC_{50}$ and sequences of modified siRNA C6G25S, C8G25S, and C10G31A. Vero E6 cells were transfected with 10, 2, 0.4, 0.08, or 0.016 nM of each modified siRNA and challenged with virus at MOI of 0.1. Plaque-forming virions were detected 24 h after infection. The sequences and chemical modifications of sense (S) and antisense (AS) strand of each siRNA are presented with 2'-F and 2'-OMe modifications as green and black squares, respectively. No modified RNA residues and phosphorothioate interlinkages are depicted as white squares and red dots, respectively.

E $IC_{50}$ of C6 and the fully modified C6G25S. Vero E6 cells were transfected with 10, 2, 0.4, 0.08, or 0.016 nM of C6 or C6G25S before virus infection at an MOI of 0.1. The viral RNA was quantitated by RT-qPCR at 24 h after infection.

F $IC_{50}$ data for viral $RdRp$ inhibition by C6G25S. Vero E6 cells were transfected with 10, 2, 0.4, 0.08, or 0.016 nM of C6G25S before virus infection at an MOI of 0.1. The viral RNA was quantitated by RT-qPCR at 24 h after infection.

Data information: Data were analyzed with GraphPad Prism 5 software and presented as mean ± SD of three biological replicates in (B–F).

**Table 1. siRNA candidates against SARS-CoV-2.**

| siRNA | Strand | Sequence | Target genes | SARS-CoV2 strains' coverage rate | Second structure prediction | Numbers of predicted off-target genes |
|---|---|---|---|---|---|---|
| C1 | AS | UAAGAUGUUGACGUGCCUCUU | Leader | 99.60% | Weak | 36 |
| | S | GAGGCACGUCAACAUCUUAA | | | | |
| C2 | AS | UUAGUGUGAUUUAAUGCUGUU | PLP | 99.30% | None | 15 |
| | S | CAGCAUUAAAUCACACUAA | | | | |
| C3 | AS | AAACACGGUUUAAACACCGUU | PLP | 99.60% | Weak | 11 |
| | S | CGGUGUUUAAACCGUGUUU | | | | |
| C4 | AS | UUAAGUGUAGUUGUACCACUU | CLPro | 99.80% | Weak | 20 |
| | S | GUGGUACAACUACACUUAA | | | | |
| C5 | AS | AAACUACGUCAUCAAGCCAUU | CLPro | 99.70% | None | 3 |
| | S | UGGCUUGAUGACGUAGUUU | | | | |
| **C6** | **AS** | **AAAUUACCGGGUUUGACAGUU** | **RDRP** | **99.80%** | **None** | **9** |
| | **S** | **CUGUCAAACCCGGUAAUUU** | | | | |
| **C7** | **AS** | **UUAACAUAUAGUGAACCGCUU** | **RDRP** | **99.90%** | **Weak** | **23** |
| | **S** | **GCGGUUCACUAUAUGUUAA** | | | | |
| **C8** | **AS** | **UUGACUAGAGACUAGUGGCUU** | **Spike** | **99.20%** | **None** | **16** |
| | **S** | **GCCACUAGUCUCUAGUCAA** | | | | |
| C9 | AS | UAAACACGCCAAGUAGGAGUU | Spike | 99.90% | Weak | 13 |
| | S | CUCCUACUUGGCGUGUUUA | | | | |
| **C10** | **AS** | **UCUUAGUUAGCAAUGUGCGUU** | **Helicase** | **99.70%** | **None** | **14** |
| | **S** | **CGCACAUUGCUAACUAAGA** | | | | |
| C11 | AS | UUAACUAUUAACGUACCUGUU | Envelope | 99.90% | Weak | 12 |
| | S | CAGGUACGUUAAUAGUUAA | | | | |

Eleven candidate siRNAs were selected and labeled as C1–C11. The start and end sites of siRNA binding sites, and the located genes directly targeted by siRNA candidates are based on the reference genome of SARS-CoV-2, NC_045512.2. Coverage rates were calculated using the 29,871 full-genome SARS-CoV-2 sequences from the Global Initiative on Sharing All Influenza Data (GISAID) website. For the secondary structure prediction, the target site confirmed as a nonstructured area was labeled as none (Lan et al, 2020; Rangan et al, 2020). Those sites with an RNAz $P$ < 0.9 were predicted to have propensity to form secondary structures and labeled as weak. Candidates selected for high anti-SARS-CoV2 efficacy were labeled in bold letters.

C6G25S is capable of inhibiting significantly a variety of SARS-CoV-2 variants, with $IC_{50}$ of 0.46 nM for the Alpha variant, 0.5 nM for Gamma, 0.09 nM for Delta, and 0.73 nM for Epsilon variant. These data proved C6G25S, which targets the highly conserved region of viral $RdRp$ gene, is a highly effective agent to suppress multiple strains of SARS-CoV-2.

**Table 2. Genome-wide off-target evaluation via RNA-seq and subsequent RT-qPCR confirmation.**

| Gene name | Gene description | Expression level (C6G25S/Con) | Inhibition % | Knockdown efficacy% (RT-qPCR) |
|---|---|---|---|---|
| CXCL5 | C-X-C motif chemokine ligand 5 | 0.304 | 70 | 91 |
| PRMT6 | protein arginine methyltransferase 6 | 0.307 | 69 | 56 |
| REEP3 | receptor accessory protein 3 | 0.358 | 64 | 66 |
| SGPP1 | sphingosine-1-phosphate phosphatase 1 | 0.38 | 62 | 72 |
| PI4K2B | phosphatidylinositol 4-kinase type 2 beta | 0.381 | 62 | 67 |
| FEM1B | fem-1 homolog B | 0.387 | 61 | 55 |
| TRIQK | triple QxxK/R motif containing | 0.407 | 59 | 54 |
| PA2G4 | proliferation-associated 2G4 | 0.419 | 58 | 57 |
| SEC23A | SEC23 homolog A, COPII coat complex component | 0.427 | 57 | 59 |
| CLNS1A | chloride nucleotide-sensitive channel 1A | 0.441 | 56 | 65 |
| ERO1A | endoplasmic reticulum oxidoreductase 1 alpha | 0.443 | 56 | nd |
| SDC2 | syndecan 2 | 0.45 | 55 | nd |
| BLMH | bleomycin hydrolase | 0.455 | 55 | nd |
| STYX | serine/threonine/tyrosine-interacting protein | 0.456 | 54 | nd |
| GALNT7 | polypeptide *N*-acetylgalactosaminyltransferase 7 | 0.481 | 52 | nd |
| BET1 | Bet1 golgi vesicular membrane-trafficking protein | 0.483 | 52 | nd |
| SDE2 | SDE2 telomere maintenance homolog | 0.486 | 51 | nd |
| ZNF460 | zinc finger protein 460 | 0.493 | 51 | nd |
| CLIC4 | chloride intracellular channel 4 | 0.494 | 51 | nd |
| MAPRE2 | microtubule-associated protein RP/EB family member 2 | 0.499 | 50 | nd |
| RAB12 | RAB12, member RAS oncogene family | 0.5 | 50 | nd |

Downregulated genes with fold change ≥ 2 in C6G25S-treated Beas-2B cells (10 nM C6G25S) compared with no siRNA control are listed and presented with expression level and inhibition % based on the expression level. The mRNA level was confirmed by RT-qPCR and normalized to the GAPDH reference gene. ND, not determined.

### *In vivo* evaluation of pulmonary administration route of C6G25S

As peripheral blood mononuclear cells (PBMCs) did not show any significant inflammatory responses (Appendix Fig S3) to treatment with C6G25S, and because others have successfully delivered naked siRNA to the lung via intranasal instillation (IN) or aerosol inhalation (AI), we decided to use a naked siRNA approach rather than a more complex delivery system, such as virus-like particles, lipid nanoparticles, or cell-penetrating peptides that have been reported to cause adverse immune stimulation or cytotoxicity (Vangasseri *et al*, 2006; Wilson, 2009; Slütter *et al*, 2011; Farkhani *et al*, 2016). Furthermore, naked siRNA delivery via IN or AI has been widely applied to knockdown a specific gene or inhibit viral infection in lungs of different animal species (Bitko *et al*, 2005; Zafra *et al*, 2014; Kandil & Merkel, 2019), including nonhuman primates (Li *et al*, 2005) and humans (DeVincenzo *et al*, 2010; Gottlieb *et al*, 2016). To assess whether IN or AI can provide even distribution of C6G25S to the lungs, mice exposed to C6G25S by either IN or AI were sacrificed humanely and their lungs were collected for *in situ* hybridization (ISH) with a C6G25S-specific probe. The hybridization signal showed that C6G25S was evenly distributed throughout the bronchi, bronchioles, and alveoli of mice in the AI group (Fig 3A), whereas uneven distribution was observed in the lungs of mice in the IN group (Fig 3B). Lung from mice without C6G25S treatment served as a negative control (Fig 3C). Moreover, there were twice as many C6G25S probe-stained positive cells in the AI group compared with that of the IN group (Fig 3D). These data indicated that aerosol inhalation can distribute C6G25S more evenly and efficiently throughout the whole lungs than intranasal instillation.

After C6G25S was nebulized, we collected the condensed aerosol and measured C6G25S by OD260. The concentration of C6G25S was the same before and after nebulization (Fig EV1A). The integrity of the siRNA, detected via HPLC, was not affected by nebulization (Fig EV1B). The effectiveness of the siRNA after nebulization was the same as that before nebulization, evaluated by the inhibition of viral envelope gene expression in Vero E6 cells (Fig EV1C). Particle size distribution of the siRNA aerosol is presented in Fig EV1D. The particle size of the siRNA aerosol generated by the nebulizer had a mass median aerodynamic diameter (MMAD) of 4.725 μm, and a geometric standard deviation (GSD) of 2.376 μm with a fine particle fraction (FPF; < 5 μm) of 51.94%. The drug recovery rate was 90%. The nebulization rate was maintained at 0.5 ml/min when the concentration of C6G25S was ≦ 30 mg/ml and reduced significantly at higher concentrations (Appendix Fig S2A). Air samples were collected from the inhalation chamber at various time points during aerosol generation to calculate the dose of C6G25S deposited by AI,

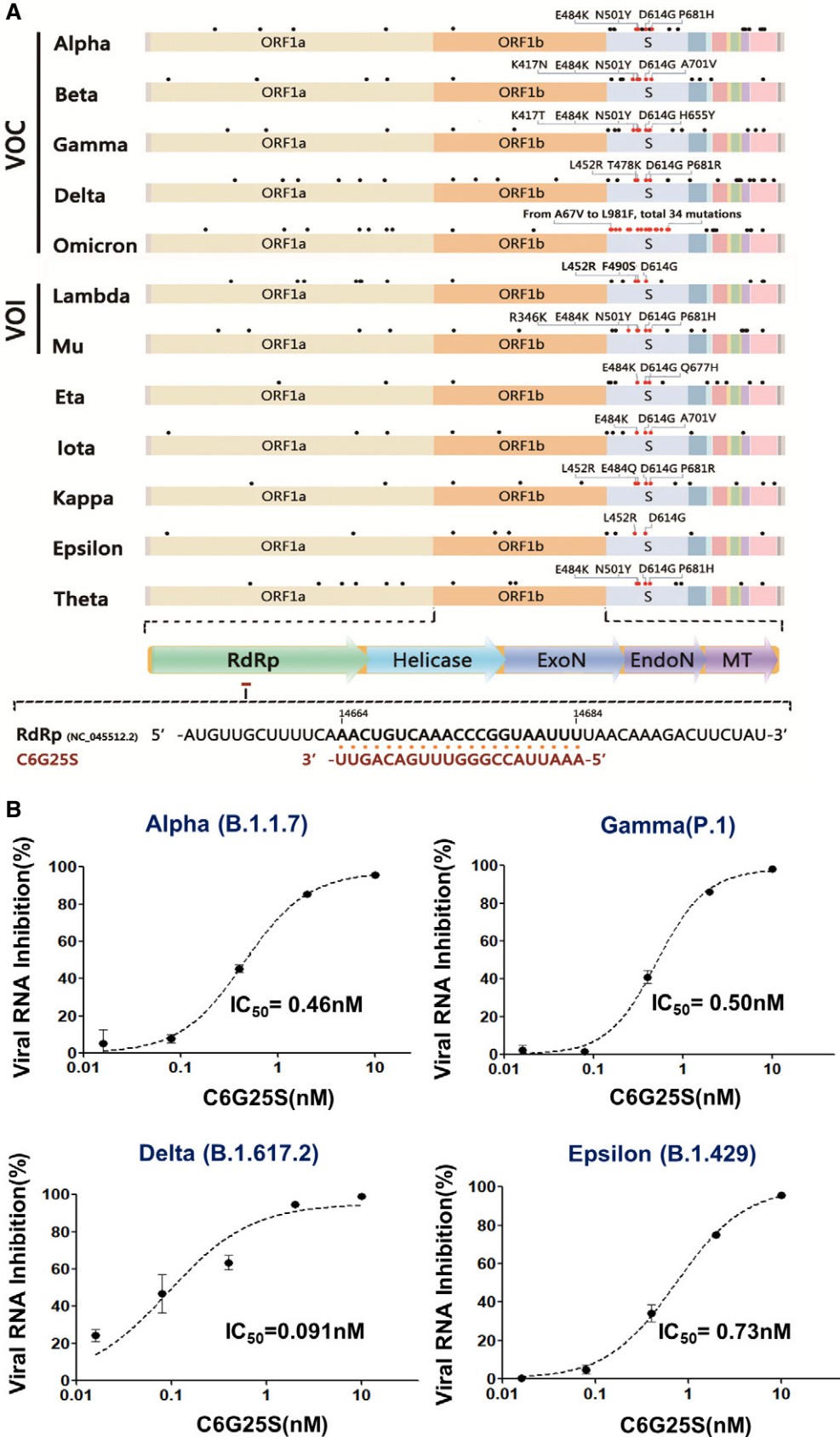

Figure 2.

**Figure 2.  C6G25S targeted and inhibited various strains of SARS-CoV-2.**

A  C6 targets a highly conserved region of the virus *RdRp* (accession number: NC_045512.2). The figure shows a genome map for five variants of concern (VOC), two variants of interests (VOI), and five other variants. Dots above the genome indicate the locations of typical mutations for each variant. Important amino acid mutations listed in the Spike mutations of interest on the European Centre for Disease Prevention and Control (ECDC) website are labeled in red. Other mutations are labeled in black. The target site and sequence for C6G25 recognition on *RdRp* is shown below the map. C6G25 sequence is shown in red and the viral sequence in black.

B  $IC_{50}$ for C6G25S against different variants. Vero E6 cells were transfected with 10, 2, 0.4, 0.08, or 0.016 nM of C6G25S before infection with different strains of SARS-CoV-2. The viral RNA was quantitated by RT-qPCR at 24 h after infection. Data are presented as mean ± SD of three biological replicates.

Source data are available online for this figure.

and the C6G25S concentration. The C6G25S concentration in the chamber reached a maximum within 2 min and was maintained at 1.48 mg/l (Appendix Fig S2B). To determine the deposition of C6G25S delivered by IN and AI, nasal cavity and whole lungs from C6G25S treated mice were collected to quantitate the distribution of C6G25S by stem–loop reverse transcription-polymerase chain reaction (RT-PCR). The C6G25S concentrations in the lung were 5.8 times higher than that in the nasal cavities when C6G25S was delivered by AI (Fig 3E). In contrast, similar concentrations were detected in the nasal cavities and lungs when C6G25S was delivered by IN, despite this, a significant variation in siRNA level in lungs was observed (Fig 3F). The elimination rate of C6G25S in lungs and nasal cavities was quantitated at different time points for mice after AI and IN treatment, and a rapid decrease in C6G25S in both nasal cavity and lung tissues was observed within 24 h (Appendix Fig S2C and D). These findings suggest a combination of IN and AI might have an advantage in achieving thorough and stable prophylactic protection.

### C6G25S significantly suppresses the production of viral RNA and infectious virions in both prophylactic treatment and cotreatment *in vivo*

To determine whether C6G25S is protective *in vivo*, K18-hACE2 transgenic mice receiving a prophylactic or cotreatment administration of C6G25S were used as an animal model. Viral quantitation at 2 days postinfection (dpi) based on previous study (Winkler *et al*, 2020) was first evaluated. Viral RNA copies were reduced by 99.95% in the prophylactic group (Fig 4A left panel) and by 96.2% in the cotreatment group (Fig 4B left panel). No plaque-forming virions were detected in the prophylactic group (Fig 4A right panel) and a significant decrease in infectious virions by 96% was observed in the cotreatment group (Fig 4B right panel). Considering that the SARS-CoV-2 Delta variant is globally pervasive and responsible for vaccine breakthrough cases, we had explored the therapeutic effects of C6G25S on this particular variant in K18-hACE2 transgenic mice. Consistent with our previous results, prophylactic treatment of the infected mice with C6G25S resulted in a 98.3% reduction of viral RNA (Fig 4C left panel) with no detectable infectious virions (Fig 4C right panel) in the lungs as compared with that of the control group. Two cotreatment groups, including two doses and three doses of C6G25S treatment after Delta variants infection, were tested. A significant inhibition of viral RNA by 72% and 88% was observed for the two-dose and the three-dose groups, respectively (Fig 4D left panel). Similar reduction in infectious virions was also noted, 90.5% for the two-dose and 92.7% for the three-dose groups (Fig 4D right panel). Our data supported that pulmonary

delivery of C6G25S possesses a strong antiviral activity *in vivo* against SARS-CoV-2 including the Delta variant in both prophylactic treatment and cotreatment.

### C6G25S inhibits spike protein expression and prevents SARS-CoV-2-induced pathological features in lungs of K18-hACE2 transgenic mice

Lungs from the infected K18-hACE2 transgenic mice without C6G25S treatment were collected and sectioned on a microtome. Immunohistochemistry demonstrated overexpression of spike proteins throughout bronchi, bronchioles, and alveoli (Fig 5A–i, ii, v, vi, ix, and x). Moreover, pathological features of COVID-19 were observed, including pneumocyte proliferation, loss of empty space in alveoli (Wang *et al*, 2020a) (Fig 5A–v), formation of syncytial multinucleated cells (Bussani *et al*, 2020) (Fig 5A–vi), and thrombosis (Bussani *et al*, 2020) (Fig 5A–x). In contrast, lung tissue from mice with prophylactic C6G25S treatment showed a significant reduction in spike protein expression and COVID-19-associated pathological features (Fig 5A–iii, iv, vii, viii, xi, and xii).

Furthermore, a significant decrease in viral RNA by ISH (stained in brown) was also observed after C6G25S prophylactic treatment (Fig 5B) with the respective viral RNA signal quantitated and shown in Fig 5F. To determine whether the SARS-CoV-2-induced infiltration of neutrophil (Wang *et al*, 2020b), lymphocyte (Puzyrenko *et al*, 2021), and macrophage (Wang *et al*, 2020a) and acute lung inflammation could be alleviated by C6G25S treatment, lung tissue from untreated and treated mice was further stained using anti-Ly6G (neutrophil; Fig 5C), anti-F4/80 (macrophage; Fig 5D), and anti-CD3 (lymphocyte; Fig 5E). Infiltration of neutrophils, macrophages, and lymphocytes was observed in the lungs of infected mice, but a significant decrease in immune cell infiltration was observed upon C6G25S treatment. The ratio of the infiltrated immune cell area to the whole section area was determined and normalized to the control group. The positively stained area of neutrophil, macrophage, and $CD3^+$ lymphocytes was reduced 78.2, 46.9, and 62.4% by C6G25S treatment, respectively (Fig 5G). The pro-inflammatory cytokines including IL-6, IFN-α, TNF-α, and IFN-γ were also analyzed. In addition to IFN-α and TNF-α that were under detection limit, the expression of IL-6 and IFN-γ was significantly reduced by C6G25S treatment (Fig EV2A). IHC staining for IL-6 (Fig EV2B) and IFN-γ (Fig EV2C) also confirmed the reduction in cytokines by C6G25S treatment. The lung injury was evaluated using the scoring system published by the American Thoracic Society in 2011 (Matute-Bello *et al*, 2011). C6G25S treatment significantly reduced SARS-CoV-2-associated lung injury in the K18-hACE2 transgenic mice (Fig 5H).

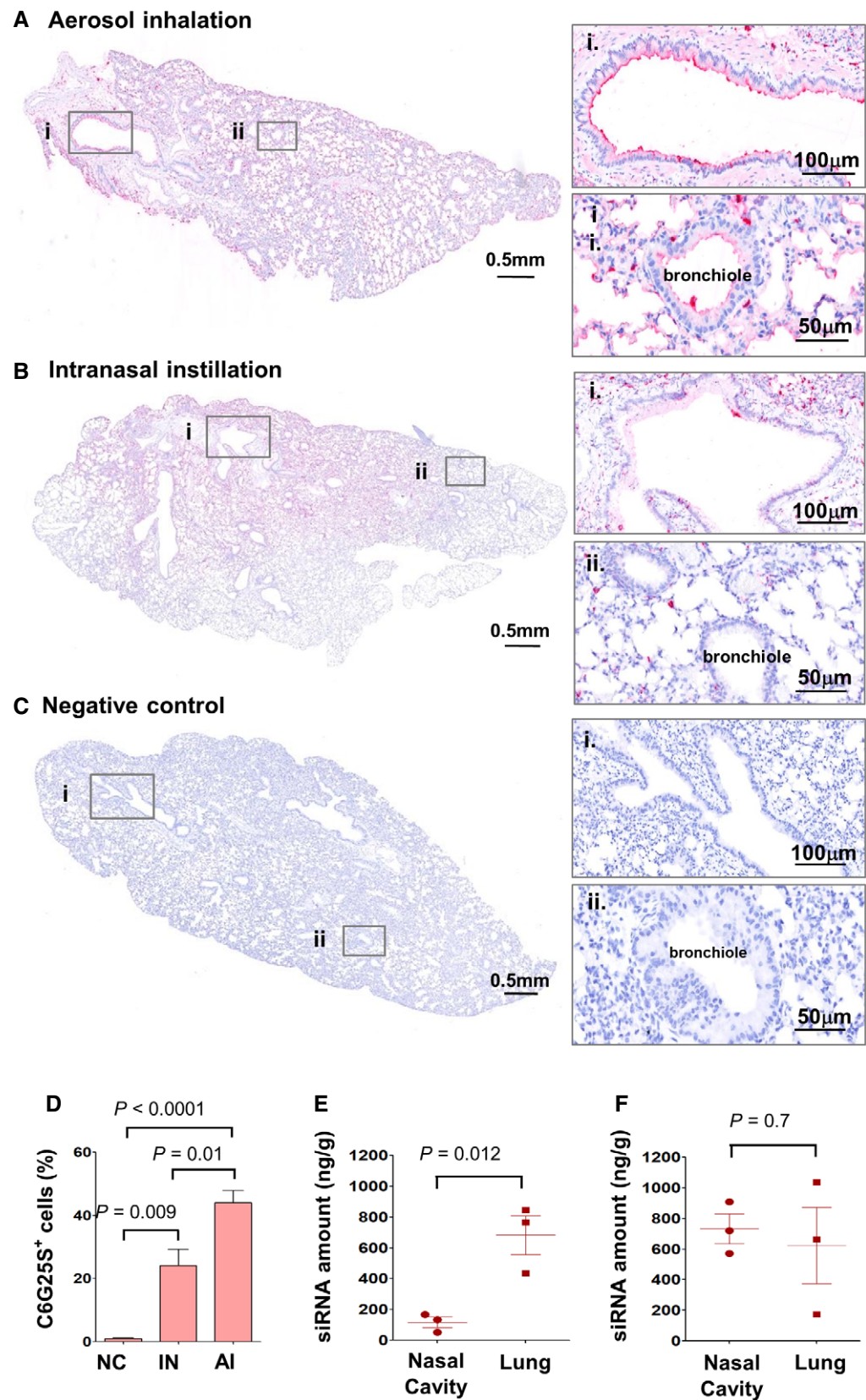

**Figure 3.**

**Figure 3. *In vivo* study of administration route for C6G25S.**

A–C    K18-hACE2-transgenic mice treated with C6G25S by 1.48 mg/l of AI for 30 min (A), 50 μl of saline containing 50 μg of C6G25S by IN (B), or PBS (C) (*n* = 5 per group). C6G25S distribution in lungs was visualized by *in situ* hybridization (ISH) staining with C6G25S-specific probe (red color). Bronchi (i) and bronchioles (ii) marked with the boxes are enlarged on the right.

D    Quantitation of C6G25S-positive cells in lungs of (A), (B), and (C). Quantitation data represent mean ± SD (*n* = 5 per group). *P*-value by Student's *t*-test.

E, F    C57/B6 mice (*n* = 3 per group) were treated by AI with C6G25S 1.48 mg/l for 30 min (E) or 50 μl of saline containing 50 μg of C6G25S by IN administration (F). The C6G25S deposited in whole lungs and nasal cavities was quantitated after whole tissue homogenization followed by stem-loop RT–qPCR. Quantitation data represent mean ± SD. *P*-value by Student's *t*-test.

Source data are available online for this figure.

### C6G25S is nonimmunogenic and well-tolerated *in vivo*

To investigate the potential local immune response of C6G25S, lungs from ICR mice treated with efficacy dose of C6G25S were analyzed for the expression of pro-inflammatory cytokines including IL-6, TNF-α, IFN-γ, and IFN-γ. C6G25S treatment did not induce any cytokine expression. In contract, *Poly*(I:*C*) induced the expression of IL-6, TNF-α, and IFN-γ and an extensively used control siRNA (Kermorgant *et al*, 2004) was shown to induce eight-fold IL-6 expression (Fig EV3A). Despite the induction of IL-6, control siRNA did not show any inhibition of the viral RNA amplification that might be evoked through nonspecific immune responses (Fig EV3B). Furthermore, administration up to 75 mg/kg of C6G25S did not induce immune cell infiltration in bronchoalveolar lavage fluid (Fig EV4A) and pro-inflammatory cytokine expression in lung tissue (Fig EV4B). To determine the clinical utility of C6G25S, human PBMCs were cocultured with 10 μM of C6G25S. No cytokines, such as interleukin (IL)-1α, IL-1β, IL-6, IL-10, tumor necrosis factor-α, nor interferon-γ were significantly induced (Appendix Fig S3). Additionally, no cytotoxicity was observed when BEAS-2B, a human cell line from normal bronchial epithelium, was exposed to a higher concentration of C6G25S in a cytotoxicity assay (Appendix Fig S4). To determine the potential adverse effects of C6G25S *in vivo*, a single-dose toxicity study with a single dose up to 75 mg/kg was conducted in Sprague–Dawley rats, and a 14-day repeated-dose study with daily dose up to 50 mg/kg was conducted in mice. In both studies, no animal death, body weight change, or drug-related adverse effect was observed within the monitoring period (Appendix Fig S5). Moreover, histopathology, hematology, and blood biochemical analysis revealed no abnormalities in either single-dose toxicity study (Appendix Tables S1–S3) or 14-day repeated-dose toxicity study (Appendix Tables S4–S6).

## Discussion

To summarize, C6G25S, a specifically designed siRNA targeting a highly conserved *RdRp* region of SARS-CoV-1/2, was demonstrated to potently inhibit infection of various SARS-CoV-2 strains through RNA interference mechanism that specifically cleaves complementary viral RNA at the C6G25S recognition site (Fig 6). IN delivery of naked siRNA has been proven to be an effective approach to prevent and treat SARS-CoV-1 infection in nonhuman primates (Rhesus macaque) (Li *et al*, 2005). Because the SARS-CoV-1 outbreak had been brought under control in a short period of time, the therapeutics has not proceeded to clinical use. Recently, several studies testing siRNA against SARS-CoV-2 infection have

been published (Niktab *et al*, 2021; Shawan *et al*, 2021; Tolksdorf *et al*, 2021; Wu & Luo, 2021). While most researchers have pursued a siRNA design that has been validated through *in vitro* cell-based experiments, only two publications have assessed the efficacy of their siRNA in an animal model. One publication used intravenous administration of LNPs-siRNA to inhibit SARS-CoV-2 infection in hACE2 transgenic mice (preprint: Idris *et al*, 2021), and the viral titer was reduced by about a log of magnitude at day 3 postinfection by prophylactic treatment. The other study used positive-charged dendrimer to carry siRNA and treated SARS-CoV-2-infected Syrian hamster via inhalation (Khaitov *et al*, 2021). The viral titer was reduced by 30% at day 2 postinfection. The potency of viral inhibition reported in these studies was not as significant as ours. Moreover, LNPs and positive charge dendrimer have been found to be capable of inducing immune or cell toxicity (Kedmi *et al*, 2010; Kharwade *et al*, 2021), which might limit the safety window for dosing. These properties also increase the difficulty and cost of industrial production. In contrast, the feasibility and safety of respiratory-delivered, unmodified naked siRNA has been demonstrated in animal models and in clinical trials (Li *et al*, 2005; DeVincenzo *et al*, 2010). Taken together, fully modified C6G25S with low immunogenicity (Figs EV3A and EV4) and reduced off-target effect (Appendix Fig S1) was developed for naked delivery to the respiratory system as a safe, effective, and feasible approach against SARS-CoV-2.

Interestingly, we found miR-2911, a natural microRNA that had been reported to inhibit SARS-CoV-2 (Zhou *et al*, 2020), had a predicted target sites overlapping with C6G25S (Appendix Fig S6A), but it was shown to only reduce 72% of original virus and was unable to inhibit alpha variant in *in vitro* assay (Appendix Fig S6B). Moreover, the C–U transversion of Alpha variants located at the ninth nucleotide in C6G25S targeting site is tolerated by siRNA recognition (Huang *et al*, 2009) and can still be inhibited by C6G25S with a IC$_{50}$ of 0.46 nM (Fig 2B, Alpha variant). This finding suggests that C6G25S is a much promising therapeutic compared to miR-2911. The coverage rate of C6G25S is 99.8% when using over 200,000 SARS-CoV-2 genome sequences downloaded from the National Center for Biotechnology Information on Aug 22, 2021 (Appendix Fig S7), and we believe that its efficacy could be further enhanced if C6G25S is combined with other highly potent siRNAs such as C8 or C10 identified in this study.

The distribution of C6G25S across the entire lung was uniform when delivered via AI, but not IN (Fig 3E). However, quantitation analyses indicated that IN has a much higher dosing efficiency within the nasal cavity (Fig 3F), which prompted us to propose a combination strategy for prophylactic treatment of hACE2-transgenic mice. An averaged 99.9% reduction in viral RNA and no

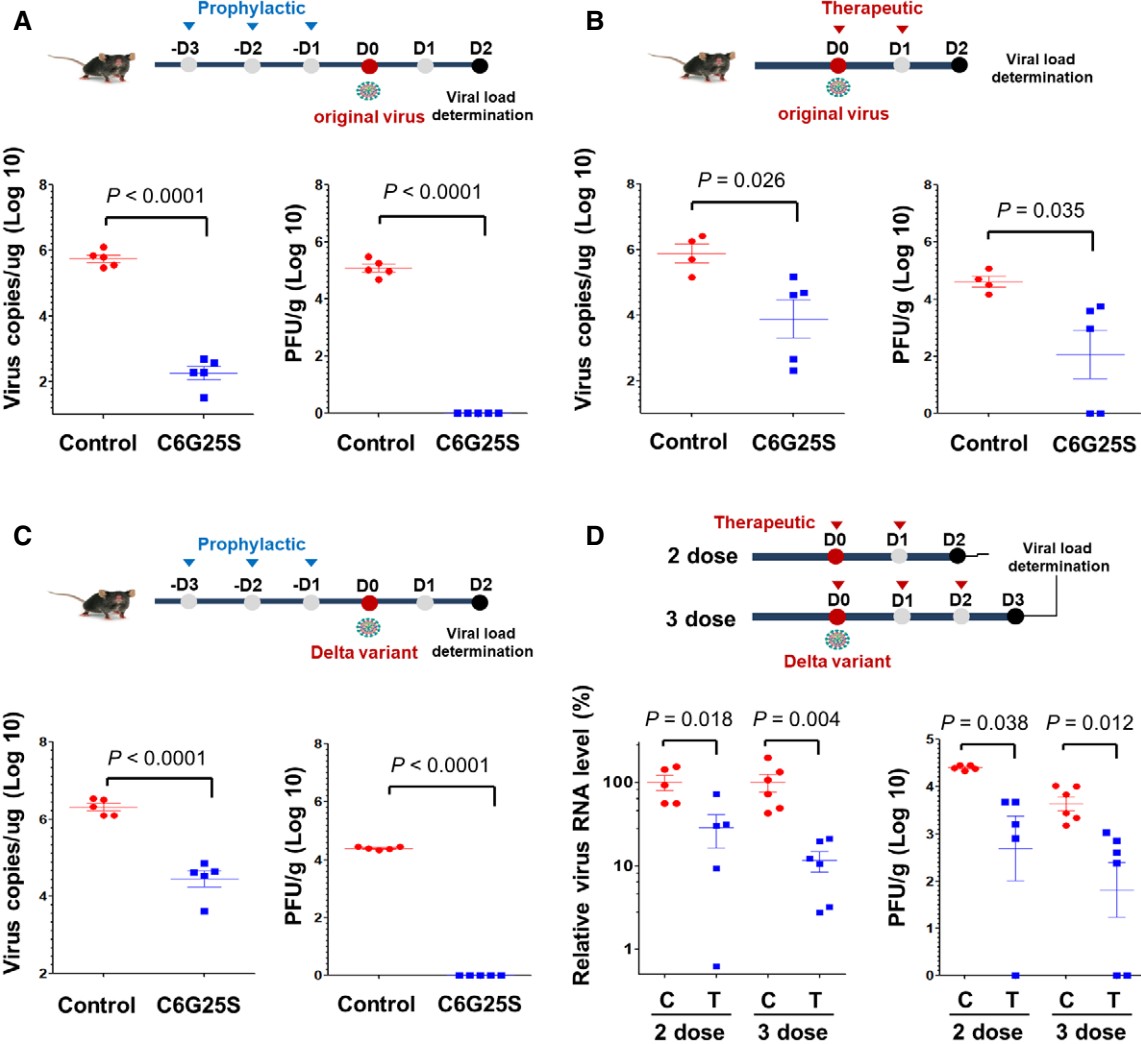

**Figure 4. Prophylatic and cotreatment administration of C6G25S in treatment of SARS-CoV-2 and Delta variant *in vivo*.**

A  K18-hACE2 transgenic mice (Winkler *et al*, 2020) were treated once daily for 3 days before intranasal challenge with $10^4$ plaque-forming units (PFU) of the original virus. Prophylactic treatment consists of 30 min of AI (1.48 mg/l of C6G25S), followed by IN of 50 μg C6G25S. Mice receiving vehicle control (saline) for both AI and IN are annotated as control. Viral RNA (left) and infectious virions (right) in lungs were quantitated with RT-qPCR and plaque forming assay, respectively, at 2 days postinfection (dpi).

B  Mice were challenged intranasally with $10^4$ PFU of virus and cotreatment with 1.48 mg/l of C6G25S or vehicle control (saline) by AI for 30 min on day 0 (right after infection) and day 1. Viral RNA and infectious virions were quantitated at 2 dpi.

C  Prophylactic treatment against Delta virus with the same experimental design as in (A). Viral RNA (left) and infectious virions (right) in lungs were quantitated at 2 dpi.

D  Cotreatment of C6G25S against Delta virus. The two-dose group was treated at day 0 and day 1, and analyzed at day 2 dpi. The three-dose group was treated at day 0, day 1, and day 2, and then analyzed at day 3 dpi. Virus RNA level was assessed relative to controls of each time point. The treatment group is labeled as T and the vehicle control is labeled as C.

Data information: Data are presented as mean ± SD. *P*-value by Student's *t*-test.

Source data are available online for this figure.

measurable plaque-forming virions were detected after the prophylactic treatment (Fig 4A). In the cotreatment, viral RNA was reduced by 96.2% and infectious virions by 96.1% via inhalation (Fig 4B). Moreover, spike protein expression and immune cell infiltration in the lungs of infected mice receiving C6G25S treatment were both significantly decreased, along with reductions in disease-associated pathological features (Fig 5).

Furthermore, one of the major off-target genes of C6G25S, *CXCL5*, is a chemotactic factor secreted by lung epithelial cells and has a participatory role in COVID-19-associated pathogenesis by induction of neutrophil infiltration and acute lung injury (Nouailles *et al*, 2014; Tomar *et al*, 2020). These findings suggest that C6G25S might have a unique dual effect that can simultaneously inhibit SARS-CoV-2 infection and reduce the risk of severe illness.

In conclusion, the promising efficacy of respiratory-delivered naked C6G25S has been demonstrated in this report. However, in spite of picomolar range of $IC_{50}$ *in vitro*, a relatively high administered dose of C6G25S is still needed. If a safe and effective delivery system can be employed, C6G25S has a significant potential to improve the efficacy and reduce medication dosage. Precision therapy is likely to be further improved by targeting specific ligands on infected cells. Giving the fact that the vaccine-resistant variants continue to increase the uncertainty in the battle against the pandemic, the siRNA strategy using the C6G25S motif, can be a

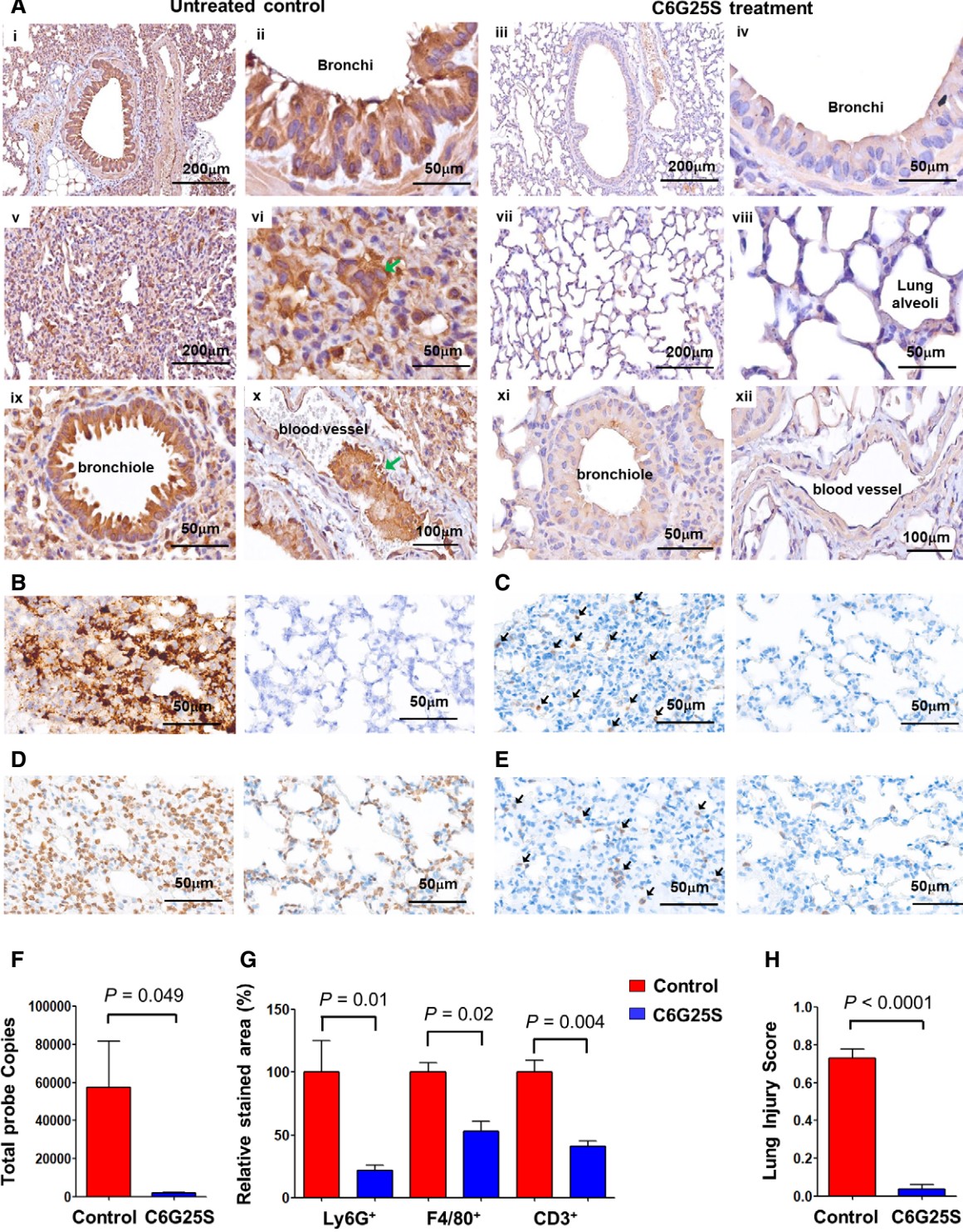

**Figure 5.**

**Figure 5.  C6G25S prevents SARS-CoV-2-induced tissue damage in the lungs of K18-hACE2 transgenic mice.**

A   Immunohistochemical (IHC) staining of viral spike proteins in lung sections from K18-hACE2 mice (Winkler *et al*, 2020) at day 5 postinfection. Spike proteins were detected with anti-spike antibody and stained brown. Images of bronchial epithelium of vehicle control-treated (i and ii) and C6G25S-treated (iii and iv), alveoli of vehicle control-treated (v and vi) and C6G25S-treated (vii and viii), bronchiole and blood vessel of vehicle control-treated (ix and x), and C6G25S-treated (xi and xii) groups were shown. Syncytial cell was indicated by green arrow in (xi) and thrombosis in (x).

B   ISH staining of viral RNA in lungs of vehicle control group (left) and C6G25S-treated group (right) at day 2 postinfection. Viral RNA was stained brown as green arrows indicated. Images are representative of five animals for each group.

C   Images of IHC staining of Ly6G$^+$ Cells (brown color, arrows indicated) in lungs of vehicle control (left) and C6G25S-treated group (right) at day 2 postinfection.

D   Images of IHC staining of F4/80$^+$ Cells (brown color) in lungs of vehicle control (left) and C6G25S-treated group (right) at day 2 postinfection.

E   Images of IHC staining of CD3$^+$ T Cells (brown color, arrows indicated) in lungs of vehicle control (left) and C6G25S-treated group (right) at day 2 postinfection.

F   Quantitative analysis of ISH images from (B). Whole lung section per mouse and five mice per group (vehicle control-treated and C6G25S-treated group) were measured. Data represent mean ± SD, *P*-value by Student's *t*-test.

G   Quantitative analysis of lung-infiltrated immune cells in whole sections of (C), (D), and (E). Percentage of positively stained areas for vehicle control group (*n* = 5) was measured and normalized to 100. The relative percentage of positively stained areas for vehicle control-treated group was shown in red and C6G25S-treated group shown in blue (*n* = 5). Data represent mean ± SD, *P*-value by Student's *t*-test.

H   Lung injury scores were calculated for five mice per groups. Data represent mean ± SD, *P*-value by Student's *t*-test.

Source data are available online for this figure.

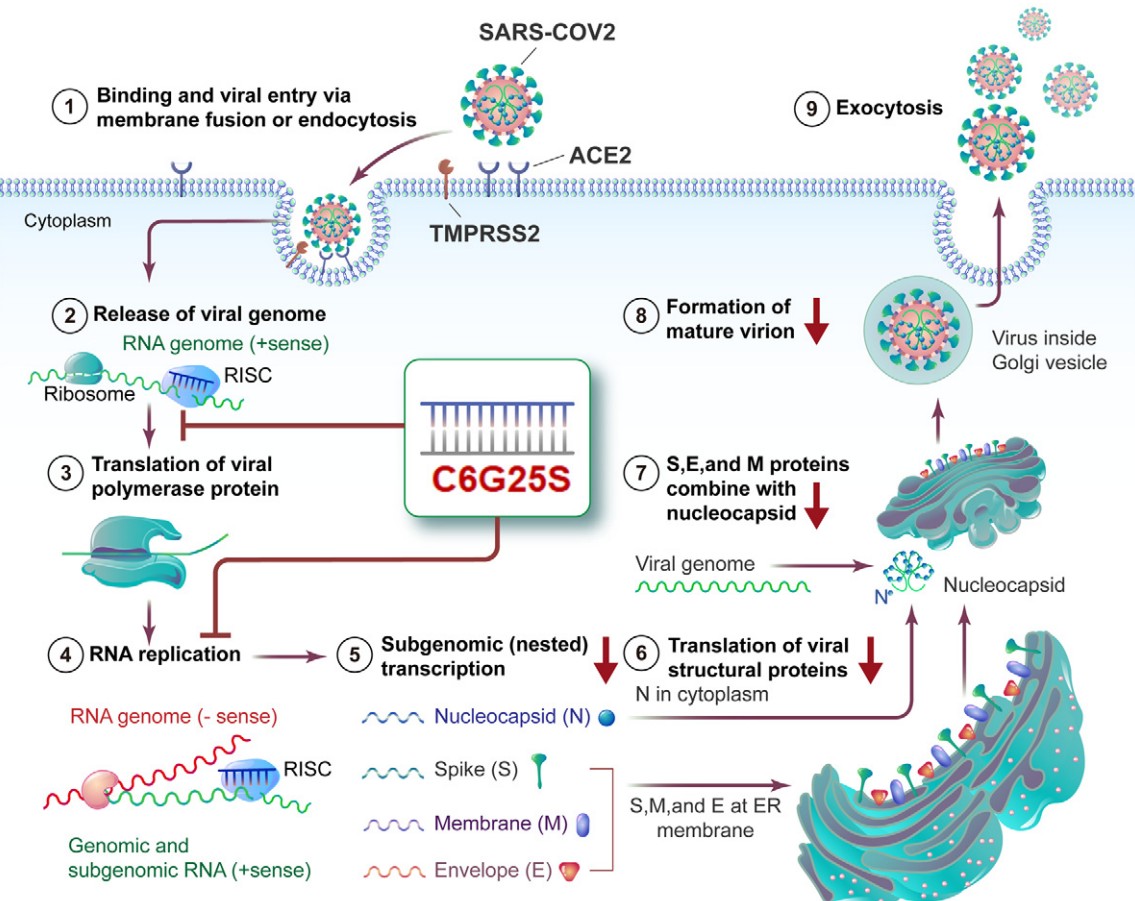

**Figure 6.  Flowchart for the mechanism of action of C6G25S.**

SARS-CoV-2 binds to ACE2 receptors on the host cell and induces endocytosis. Cleavage of the viral spike protein by TMPRSS2 triggers membrane fusion and subsequent release of the viral sense (+) RNA genome. After hijacking the host's ribosome, viral RNA-dependent RNA polymerase is generated to facilitate further virus replication. Meanwhile, subgenomic transcription and translation generate large amounts of viral structural proteins, such as the nucleocapsid, spike, membrane, and envelope. The progeny virus is assembled and the mature virions are released by exocytosis. C6G25S can interact with the RNA-induced silencing complex to digest the viral genome's RNA and polymerase mRNA through the RNAi effect. By reducing the copy number of the viral genome and polymerase mRNA, the virus replication cycle are inhibited, the productive SARS-CoV-2 infection is interrupted.

promising therapeutic to tackle and inhibit SARS-CoV-2 variants and may address the bottleneck of current therapies for COVID-19 with a good safety profile.

# Materials and Methods

### SARS-CoV-2-specific siRNA selection

As of June 2020, there were 29,871 full-length SARS-CoV-2 genome sequences available from the Global Initiative on Sharing All Influenza Data (GISAID) website. These sequences were analyzed for 19-nucleotide stretches that showed at least 99% identity (high conservation) in the SARS-CoV-2 genome. Because RNA target accessibility affects the siRNA efficacy, the viral RNA secondary structure was evaluated based on RNA structure *in vivo* analyzed by genome-wide dimethyl sulfate mutational profiling with sequencing (DMS-MaPseq) (Lan *et al*, 2020) and *in silico* prediction with RNAz (RNAz *P* < 0.9) (Rangan *et al*, 2020). Sequences targeting viral regions with a strong secondary structure were removed (RNAz *P* > 0.9). A total of 674 siRNA candidates showed over 99% coverage and the targeted regions had low propensity for RNA secondary structure. We selected those candidates located within regions coding the viral leader, papain-like protease, 3C-like protease, RdRp, helicase, spike protein, and the envelope protein for further off-target prediction and essential gene targeting. Off-target effects were predicted via blast with GRCh38 reference sequence database and candidates were filtered with the number of off-target genes ≤ 36. Off-target genes were further evaluated for their essential contribution to cell viability (Blomen *et al*, 2015; Hart *et al*, 2015; Wang *et al*, 2015). Candidates were selected with a low number of essential genes predicted to be targeted by the siRNA candidates ($n \leq 1$). The top 11 siRNA candidates were identified for subsequent *in vitro* screening by viral RNA knockdown and plaque reduction assay in Vero E6 cells.

### Cells and viruses

Vero E6 cells were maintained in Dulbecco's modified Eagle's medium (DMEM) supplemented with 10% fetal bovine serum (FBS) at 37°C with 5% $CO_2$. The human bronchial epithelial cell line BEAS-2B was maintained in RPMI-1640 medium supplemented with 10% FBS. Sputum specimens obtained from patients infected with SARS-CoV-2 were maintained in viral transport medium. The virus in the specimens was propagated in Vero E6 cells in DMEM supplemented with 2 μg/ml tosylsulfonyl phenylalanyl chloromethyl ketone (TPCK)-trypsin (Sigma-Aldrich). Culture supernatant was harvested when cytopathic effect (CPE) was observed in more than 70% of cells, and virus titers were determined by plaque assay. The virus isolates used in *in vitro* siRNA screening and IC$_{50}$ determination were hCoV-19/Taiwan/NTU13/2020 (A.3; EPI_ISL_422415), hCoV-19/Taiwan/NTU49/2020 (B.1.1.7; EPI_ISL_1010728), hCoV-19/Taiwan/NTU56/2021 (B.1.429; EPI_ISL_1020315), hCoV-19/Taiwan/CGMH-CGU-53/2021 (P.1; EPI_ISL_2249499), hCoV19/Taiwan/NTU92/2021 (B.1.617.2; EPI_ISL_3979387). The viruses used in the infection of K18-hACE2 transgenic mice were hCoV-19/Taiwan/4/2020 (B; EPI_ISL_411927) and hCoV-19/Taiwan/1144/2021 (B.1.617.2; EPI_ISL_5854263).

### siRNA screening in Vero E6 cells

Vero E6 cells were resuspended in culture medium at $2 \times 10^5$ cells/ml and reverse-transfected with siRNA as follows: siRNA and Lipofectamine RNAiMAX (Thermo Fisher Scientific) were diluted with Opti-MEM I reduced serum medium (Thermo Fisher Scientific/Gibco) separately. The siRNA/Opti-MEM mixtures were added to the Lipofectamine RNAiMax/Opti-MEM mixtures. The siRNA-RNAiMAX mixtures (100 μl) were incubated for 10 min at room temperature. Vero E6 cells (500 μl, $2 \times 10^5$ cells/ml) were then added to the siRNA-RNAiMAX mixtures and transferred into a 24-well plate. Negative control siRNA (NC siRNA) was used as a transfection control (Kermorgant *et al*, 2004). The sense and antisense sequence of NC siRNA was 5′-UUC UCC GAA CGU GUC ACG UTT-3′ and 5′-ACG UGA CAC GUU CGG AGA ATT-3′.

After 24 h incubation, the siRNA-transfected Vero E6 cells were infected with SARS-CoV-2 virus at a multiplicity of infection (MOI) of 0.1. After 1 h incubation, the inoculum was removed and the cells were washed with phosphate-buffered saline (PBS). Fresh medium was added for incubation at 37°C for 24 h. After that, culture supernatant was harvested for plaque assay (Cheng *et al*, 2020) and the total cellular RNA was extracted with a NucleoSpin RNA mini kit (Macherey–Nagel) to determine the amount of viral RNA by reverse transcription-quantitative polymerase chain reaction (RT-qPCR) of viral E gene on a QuantStudio 5 Real-Time PCR System (Applied Biosystems) using an iTaq Universal Probes One-Step RT-PCR Kit (Bio-Rad) (Cheng *et al*, 2020). The primers and probe targeting the SARS-CoV-2 were as follows: forward primer, 5′-ACA GGT ACG TTA ATA GTT AAT AGC GT-3′; reverse primer, 5′-ACA TTG CAG TAC GCA CAC A-3′; and probe, 5′-ACA CTA GCC ATC CTT ACT GCG CTT CG-3′. Plasmid containing partial E fragment was used as a standard to calculate the viral load (copies/μl). All work involving SARS-CoV-2 virus was performed in a Biosafety Level-3 Laboratory at the National Taiwan University College of Medicine with approval from the institutional Biosafety Committee.

### siRNA delivery via inhalation and intranasal instillation

Inhalation delivery of siRNA was performed using a standard device consisting of a polycarbonate chamber connected to a Aeroneb Lab Nebulizer Unit at 0.5 ml/min. Mice ($n = 5$/group) were placed into the chamber and aerosol was generated for 25 min from 10 ml normal saline containing 6 mg/ml siRNA or 10 ml saline alone (control) for prophylactic treatment or 12 mg/ml siRNA for cotreatment. Mice were exposed to siRNA aerosol or control saline aerosol in the chamber for a total of 30 min. For intranasal administration, 50 μg siRNA in 50 μl of D5W or D5W alone (control) was instilled into both nostrils (25 μl per nostril). To compare the difference between inhalation and intranasal instillation on the distribution and concentration of siRNA in the lungs and nasal cavities, K18-hACE2 transgenic mice or C57BL/6 mice were treated with siRNA aerosol generated from 10 ml normal saline containing 6 mg/ml siRNA. Intranasal instillation was performed with 50 μg siRNA in 50 μl of D5W. Control group mice were administered with vehicle alone. The distribution of siRNA in lungs was analyzed via *in situ* hybridization. The siRNA level in lungs and nasal cavities was quantitated by stem-loop qPCR using a standard curve. The siRNA concentration of the aerosol in the chamber was quantitated as

follows. Aerosol samples were collected from the chamber using 0.5 ml syringes at 1, 2, 5, 15, and 25 min after aerosol was generated, and then passed through 100 µl nuclease-free water. The siRNA level in the nuclease-free water was subsequently determined by OD260. The maximum siRNA level, $B_{max}$, was calculated and presented as mg/l air aerosol.

## Quantitation of siRNA level in lungs and nasal mucosa

C57BL/6 mice were sacrificed at different time points after pulmonary delivery or intranasal instillation of siRNA. Livers and nasal mucosa were weighed and homogenized in 0.25% Triton X-100 in PBS to a final concentration of 100 mg/ml using a TissueLyser II (Qiagen) at 4°C. siRNA level was quantitated by stem-loop RT-qPCR (Brown *et al*, 2020). Briefly, homogenized samples were heated to 95°C for 10 min, briefly vortexed, and cooled on ice for 10 min. The resultant tissue lysate was collected after centrifugation at 20,000 *g* for 20 min at 4°C. Antisense-specific cDNA was generated from tissue lysate using a stem-loop cDNA primer: 5′-GTC GTA TCC AGT GCA GGG TCC GAG GTA TTC GCA CTG GAT ACG ACA ACT GTC A-3′. qPCR was performed on a QuantStudio 6 Flex Real-Time PCR System (Thermo Fisher Scientific) using Power SYBR Green PCR Master Mix (Thermo Fisher Scientific): forward primer, 5′-AAG CGC CTA AAT TAC CGG GTT-3′; reverse primer, 5′-GTG CAG GGT CCG AGG T-3′. Antisense strand level was quantitated using a standard curve generated by spiking the synthetic siRNA into the corresponding naïve tissue matrix of the same concentration.

## *In vitro* aerodynamic deposition study

The *in vitro* aerodynamic attributes including mass median aerodynamic diameter (MMAD), geometric standard deviation (GSD), and fine particle dose, fine particle fraction (FPF) were measured at Micro-Base Technology (Taoyuan City, Taiwan) using the next-generation impactor (NGI) and a USP induction port (Copley Scientific, Nottingham, UK). The NGI was assembled and operated in accordance with USP General Chapter 1601 to assess the drug delivered.

## siRNA treatment and virus infection of K18-hACE2 mice

Eight–sixteen-week-old K18-hACE2 transgenic mice (McCray *et al*, 2007) were purchased from The Jackson Laboratory and inbred in Laboratory Animal Center of National Taiwan University College of Medicine (Taipei, Taiwan, ROC). For the prophylactic treatment, mice were treated with aerosolized siRNA or aerosolized vehicle (saline) and the subsequent intranasal instillation of siRNA or vehicle control daily for 3 days before virus infection (D-1 to D-3). Twenty-four hours after the last siRNA treatment (D0), mice were anesthetized with Zoletil/Dexdomitor and infected intranasally with $10^4$ plaque-forming units (pfu) of SARS-CoV-2 in 20 µl of DMEM, followed by Antisedan administration. For cotreatment, mice were first anesthetized with Zoletil/Dexdomitor and infected intranasally with $10^4$ PFU of SARS-CoV-2. After 30 min of recovery, mice were placed in the chamber to perform 30 min of inhalation treatment (D0). Mice were treated with aerosolized siRNA at D0 and 1 day postinfection. Infected mice were sacrificed to collect their lungs at 2 days postinfection. All work with SARS-CoV-2 was conducted in a Biosafety Level (BSL)-3 or BSL-4 Laboratories at the Institute of Preventive Medicine, the National Defense Medical Center (Taiwan, ROC) with approval from the Institutional Biosafety Committee and Institutional Animal Care and Use Committee (IACUC).

## Quantitation of SARS-CoV-2 RNA and infectious virus in lungs

Lungs were suspended in 1 ml DMEM supplemented with 1× antibiotic–antimycotic (Gibco) before further homogenization using beads in a Precellys tissue homogenizer (Bertin Technologies). Tissue homogenates were clarified by centrifugation at 12,000 *g* for 5 min at 4°C. The supernatants were collected for determination of infectious virus by plaque assay and viral RNA titers by RT-qPCR. The clarified lung homogenates were mixed with a five-fold excess of TRI reagent (Sigma-Aldrich). RNA was extracted following the manufacturer's instructions (TRI reagent). The extracted RNA was dissolved in 100 µl nuclease-free water. Viral RNA was quantitated using SensiFAST Probe No-ROX One-Step Kit (catalog No. BIO-76005, Bioline) on the LightCycler 480 (Roche Diagnostics). Primers and Probe targeting the viral E gene were purchased from Integrated DNA Technologies (catalog Nos. 10006888, 10006890, 10006893). RT-qPCR was performed with 500 ng of total RNA, 400 nM of each forward and reverse primer, and 200 nM probe in a total volume of 20 µl. The cycling conditions were as follows: 55°C for 10 min, 94°C for 3 min, and 45 cycles of 94°C for 15 s and 58°C for 30 s. The amount of viral RNA was calculated using a standard curve constructed from an RNA standard. The virus titer in the clarified lung homogenates was quantitated using a plaque assay. Briefly, Vero E6 cells ($1.5 \times 10^5$ cells/well) were seeded in 24-well tissue culture plates in DMEM supplemented with 10% fetal bovine serum (FBS) and antibiotics. The 10-fold serial diluted homogenates were inoculated into Vero E6 cells for 1 h at 37°C with shaking occasionally. After removing the supernatant, cells were washed once with PBS, overlaid with 1.55% methylcellulose in DMEM with 2% FBS, and then incubated for another 5 days. The methylcellulose overlays were removed after 5 days of incubation. Cells were fixed with 10% formaldehyde for 1 h, and stained with 0.5% crystal violet. Plaques were counted to calculate PFU/g according to lung weight.

## *In situ* hybridization

Lungs and nasal cavities were fixed in formalin, embedded in paraffin, and sectioned at 4 µm thickness. The localization of C6 siRNA was investigated in tissue sections using the miRNAscope Intro Pack HD Reagent Kit RED - Mmu (Advanced Cell Diagnostics [ACD]) according to ACD's formalin-fixed paraffin-embedded tissue protocol. The probe for the detection of C6 siRNA was custom-synthesized by ACD. The hybridization signal of C6 siRNA was visualized by Fast Red, followed by counterstaining with hematoxylin. SARS-CoV-2 RNA was detected using the RNAscope 2.5 HD Reagent Kit–Brown (ACD) and RNAscope Probe- V-nCoV-2019-S (ACD). The hybridization signal of SARS-CoV-2 RNA was visualized using a 3,3′-diaminobenzidine (DAB) reagent. RNA quality in the tissue sections was verified using the probe targeting U6 snRNA as a positive control and scrambled probe as a negative control. The whole-slide images were acquired using a Ventana DP200 slide scanner (Roche Diagnostics) and processed using HALO software (Indica Labs). Quantitative comparison of ISH signals was analyzed using the HALO software with RNAscope modules.

### Immunohistochemical analysis

Formalin-fixed, paraffin-embedded lung sections were dewaxed and rehydrated and antigen retrieval was performed with Tris-EDTA buffer (pH 9.0). Endogenous peroxidase in the sections was quenched with 3% hydrogen peroxide and the tissue immunostained using a Histofine Mousestain Kit (Nichirei Biosciences) according to the manufacturer's protocol. SARS-CoV-2 spike protein, neutrophils, macrophages, and $CD3^+$ T cells were detected by incubation with anti-SARS-CoV/ SARS-CoV-2 (COVID-19) spike antibody (1:100, Clone 1A9, GTX632604, Genetex), anti-LY-6G (1:50, Clone 1A8, 551459, BD), anti-F4/80 (1:500, Clone Rb167B3, HS-397 008, Synaptic Systems), anti-CD3 (prediluted, clone 2GV6, 790-4341, Ventana Medical Systems) in primary antibody diluent (ScyTek) at 4°C overnight. IL-6 and IFN-γ were detected using Anti-IL-6 (1:200, BS-0781R) and anti-IFN-γ (1:50, BS-0480R) purchased from Bioss Company. Sections were stained using DAB reagent, counterstained with hematoxylin, and then dehydrated and mounted under cover slips. Whole slides were scanned on a Ventana DP200 slide scanner (Roche Diagnostics) and analyzed using HALO software (Indica Labs). The severity of lung injury was assessed based on the presence of neutrophils in the alveolar space, neutrophils in the interstitial space, hyaline membranes, proteinaceous debris is filling the airspaces, and alveolar septal thickening, following the method as described by Matute-Bello *et al* (2011).

### Quantitation of cytokine mRNA via RT-qPCR

The lungs were homogenized in RLT buffer (Qiagen) using a Tissue-Lyser II (Qiagen) at 4°C and clarified by centrifugation at 15,000 *g* for 15 min at 4°C. Total RNA was extracted with a RNeasy Micro Kit (Qiagen) following the manufacturer's instructions. Reverse transcription reaction was conducted with a Maxima First-Strand cDNA Synthesis kit (Thermo Fisher Scientific) and 1 ug of total cellular RNA. qPCR was carried out on a LightCycler 480 using SYBR Green I Master (Roche Diagnostics) with 1:5 dilutions of cDNA. Primer sets used to detect cytokine genes are shown in Appendix Table S7. Each sample was assayed in triplicate to determine an average threshold cycle ($C_t$) value. Gene expression fold change was calculated using the $\Delta\Delta C_t$ method. The mRNA level of each gene was normalized to constitutively expressed GAPDH mRNA.

### *In vitro* and *in vivo* evaluation of immunogenicity of C6G25S

The PBMC from healthy donors were obtained from StemExpress with Institutional Review Board (IRB) approval (IRB No. 20152869). PBMCs were resuspended in RPMI 1640 medium supplemented with 10% FBS. A total of $1 \times 10^5$ viable PBMC were added to each well of a 96-well culture plate. After 4 h, cells were treated with different concentrations 10 μM of siRNA, 1 μM CpG, and 100 μg/ml poly(I:C) (Sigma) for 40 h. Concentrations of cytokine IL-1α, IL-1β, IL-6, IL-10, TNF-α, and IFN-γ in the supernatant were quantitated using the Cytometric Bead Assay (CBA) Flex Set (BD Biosciences) according to the manufacturer's instructions. The data were collected on a FACS LSRFortessa flow cytometer (BD Biosciences) and analyzed using FCAP Array Software (version 3.0, BD Biosciences).

To investigate if C6G25S may cause acute local immune response in the lungs, 7-week-old male Bltw:CD1(ICR) mice obtained from

Shanghai Model Organisms Center (Shanghai, China) were intranasally instilled with vehicle alone (saline), poly IC at 2.5 mg/kg (positive control), or C6G25S at 0, 20, 40, or 75 mg/kg. After 48 h, mice were euthanized by isoflurane inhalation and bronchoalveolar lavage fluid (BALF) collected via intratracheal infusion of 0.8 ml of PBS. BALF was obtained by retracting the piston of the syringe three times and centrifuged at 450 *g* for 5 min at 4 °C. The resulting cell pellet was resuspended in 200 μl of PBS and analyzed immediately using an automated hematocytometer (BX3010, Sysmex). The supernatant was stored at −80°C. The mRNA levels of cytokines in lungs were determined via RT-qPCR. NC siRNA (Kermorgant *et al*, 2004) and NC siRNA2 were used as controls. NC siRNA2 was designed by blasting the database for no match to human sequences and SARS-CoV-2 genome. The sense and antisense sequence of NC siRNA was 5′-UUC UCC GAA CGU GUC ACG UTT-3′ and 5′-ACG UGA CAC GUU CGG AGA ATT-3′. NC siRNA2 was 5′-UUC GAC CGG UAU AUG GUA GTT-3′ and 5′-CUA CCA UAU ACC GGU CGA ATT-3′.

### Genome-wide off-target analysis using RNA-seq

Beas-2B cells were seeded at $5 \times 10^5$ cell/well into 6-well culture plates and incubated for 18 h. siRNA (10 nM) was then transfected into Beas-2B cells using Lipofectamine RNAiMAX (9 μl/well, Thermo Fisher Scientific) following the manufacturer's protocol. After 24-h transfection, cells were washed twice with 1× Dulbecco's PBS and solubilized in TRIzol reagent (Thermo Fisher Scientific). Total RNA was extracted following the manufacturer's instructions and treated with DNase to avoid genomic DNA contamination. Purity ($A_{260}/A_{280}$ and $A_{260}/A_{230}$ ratios) and quality (RIN ≥ 8.0) of the extracted RNA were determined using a NanoDrop 2000 spectrophotometer (Thermo Scientific) and an Agilent 2100 bioanalyzer (Agilent Technologies, Santa Clara, CA). Quality of all extracted RNA samples was $A_{260}/A_{280} \geq 1.9$, $A_{260}/A_{230} \geq 2$, and RIN = 10.0. RNA-seq Libraries was prepared for two biological replicates using TruSeq Stranded Total RNA Library Prep Gold (Illumina) and sequenced on the NovaSeq 6000 sequencer (Illumina) according to the manufacturers' instructions. Average of 81 million reads per sample was obtained from 2× 150-bp paired-end sequencing. Raw RNA reads were filtered with minimal mean quality scores of 20 using SeqPrep and Sickle. Filtered reads were aligned to the human genome (GRCh.38.p13) using HISAT2 and then assembled using StringTie. The gene expression level was qualified by RSEM and normalized by transcripts per million (). Differentially expressed genes were identified as those with at least two-fold difference between siRNA-treated and no siRNA-treated groups using the DESeq2 package (Version 1.10.1) at the Benjamini–Hochberg adjusted *P* value ≤ 0.05. Off-target gene profile was evaluated from the number and possible cellular impact of downregulated genes. The expression level of downregulated genes was further confirmed by RT-qPCR. First-strand cDNA was synthesized with Maxima First-Strand cDNA Synthesis kit (Thermo Fisher Scientific) and 2 μg of total cellular RNA. qPCR was carried out on a LightCycler 480 (Roche Diagnostics) using SYBR Green I Master (Roche Diagnostics). Each sample was assayed in triplicate to determine an average threshold cycle ($C_t$) value. Gene expression fold change was calculated using the $\Delta\Delta C_t$ method. The mRNA level of each gene was normalized to constitutively expressed GAPDH mRNA.

## Acute and repeated-dose toxicity studies

To assess the acute toxicity of C6G25S, male Sprague–Dawley rats were obtained from BioLASCO (Taipei, Taiwan, ROC). Vehicle alone (D5W) or C6G25S in D5W was administered to 7-week-old rats (three per group) via intranasal instillation at 0, 20, 40, or 75 mg/kg at a dose volume of 0.42 ml/kg. The rats were then observed for 7 days. Repeated-dose toxicity was conducted on male Bltw:CD1 (ICR) mice obtained from BioLASCO). Vehicle alone (D5W) or C6G25S (2, 10, or 50 mg/kg) was intranasally instilled (1.67 ml/kg) to 8-week-old mice (three per group) daily for 14 days. During the study, the body weight, food consumption, and general status of the animals were monitored. At the end of each study, organs and peripheral blood were collected. Acute toxicity in rats was assessed based on the clinical signs, body weight and food consumption, hematology, blood biochemistry, and microscopic pathology of the nasal cavity and lung. The assessment of repeated-dose toxicity also included the microscopic pathology of the heart, liver, spleen, and kidney.

## CCK-8 cytotoxicity assay

Beas-2B cells were seeded at $1.77 \times 10^4$ cell/well into 96-well culture plates and incubated for 18 h. Cells were then treated with various concentrations of C6G25S (40, 20, 10, 5, and 0 μM) in triplicate for 24 h. CCK-8 solution (10 μl) was added to each well, and cells were incubated for another 3 h. Medium only with CCK-8 solution and medium without C6G25S served as blank and normal control, respectively. The absorbance at 450 nm was measured with a Multiskan Sky Microplate Spectrophotometer (Thermo Fisher Scientific). The relative cell viability/cytotoxicity was calculated according to the manufacturer's instructions.

## HPLC conditions for the determination of siRNA

The determination of siRNA was performed with a Waters ACQUITY Arc HPLC system equipped with a PDA detector. Chromatographic separation was achieved on Shodex KW-802.5 (8.0 mm I.D. × 300 mm) column and potassium phosphate buffer (3 mM sodium phosphate, 150 mM sodium chloride, 1.05 mM potassium phosphate, pH 7.4) as a mobile phase. The HPLC conditions included a column temperature at 40°C, flow rate of 1.0 ml/min, UV detection at 260 nM, injection volume of 10 μl, and run time of 15 min. C6G25S sense strand ($n$, full length), C6G25S antisense strand ($n$, full length), C6G25S sense strand with 5 bases truncated ($n$-5), C6G25S antisense strand with 5 bases truncated ($n$-5), C6G25S double strand ($n$, full length), and C6G25S double strand with 5 bases truncated in each strand ($n$-5, fully complementary) were used as controls.

## Statistical analysis

All animals in this study were randomly assigned to each vehicle control and experimental groups using manual methods not randomization tools. Numbers of animals were determined based on prior experience with the model and are provided in the figure legends. All cell experiments were conducted in three biological replicates. Results were analyzed with GraphPad Prism 5 software

### The paper explained

**Problem**

New SARS-CoV-2 variants, especially Delta, are spreading rapidly across the globe. The arise of breakthrough cases suggests that current vaccine strategy is insufficient to restrain the transmission, and more effective and safe therapeutics targeting a wide spectrum of virus variants are urgently needed.

**Results**

In this study, we showed that an inhaled and broad-spectrum siRNA molecule, C6G25S, suppresses SARS-CoV-2 infection by either prophylactic treatment or cotreatment in hACE2 transgenic mice. Both virus and pathological features in lungs were significantly reduced in infected mice. Moreover, transcriptome-wide off-target analysis and animal toxicology studies revealed that C6G25S is highly specific and well-tolerated.

**Impact**

C6G25S is the first inhaled siRNA developed to target against a wide range of SARS-CoV-2. The broad-spectrum antiviral activities against SARS-CoV-2 variants and its great safety profile implicate that C6G25S could provide a promising therapeutic solution to halt the COVID-19 pandemics.

and presented as the mean ± SD. Statistical significance between groups was determined by using Student's *t*-test. *P* values < 0.05 were considered significant. RNA-seq analysis along with the associated statistical analysis was performed using the Majorbio Cloud Platform (https://cloud.majorbio.com) and is described in the RNA-seq section.

# Data availability

The datasets produced in this study are available in the following databases: RNA-seq: European Nucleotide Archive PRJEB50508 (https://www.ebi.ac.uk/ena/browser/view/PRJEB50508).

**Expanded View** for this article is available online.

## Acknowledgements

We would like to acknowledge the services provided by the Biosafety Level-3 Laboratory of the First Core Laboratory and the Transgenic Mouse Core Facility from National Taiwan University College of Medicine; the Biosafety Level-3 Laboratory from National Taiwan University Hospital. This work was supported by Oneness Biotech Company and grants from the Ministry of Science and Technology, Taiwan (MOST-110-2740-B-002-006, MOST109-2327-B-002-009, MOST 109-2124-M-002-012, MOST 109-0210-01-18-02, MOST 110-0210-01-22-02, 110-2314-B-002-282).

## Author contributions

**Pan-Chyr Yang:** Conceptualization; Supervision; Funding acquisition; Writing—review and editing. **Yi-Chung Chang:** Conceptualization; Data curation; Validation; Investigation; Visualization; Methodology; Writing—original draft; Project administration. **Chi-Fan Yang:** Data curation; Investigation; Methodology; Writing—original draft. **Yi-Fen Chen:** Validation; Investigation; Visualization; Methodology. **Chia-Chun Yang:** Data curation; Software; Formal

    

analysis; Validation; Visualization. **Yuan-Lin Chou:** Validation; Investigation; Visualization; Methodology. **Hung-Wen Chou:** Validation; Investigation; Visualization; Methodology. **Tein-Yao Chang:** Investigation; Methodology. **Tai-Ling Chao:** Investigation; Methodology. **Shu-Chen Hsu:** Investigation; Methodology. **Siman Ieong:** Investigation; Methodology. **Ya-Min Tsai:** Investigation; Methodology. **Ping-Cheng Liu:** Investigation; Methodology. **Yuan-Fan Chin:** Investigation; Methodology. **Jun-Tung Fang:** Investigation; Methodology. **Han-Chieh Kao:** Validation; Investigation; Methodology. **Hsuan-Ying Lu:** Validation; Investigation; Methodology. **Jia-Yu Chang:** Validation; Investigation; Methodology. **Ren-Shiuan Weng:** Investigation; Methodology. **Qian-Wen Tu:** Investigation; Methodology. **Fang-Yu Chang:** Investigation; Methodology. **Kuo-Yen Huang:** Validation. **Tong-Young Lee:** Supervision. **Sui-Yuan Chang:** Conceptualization; Supervision; Writing—review and editing.

## Disclosure and competing interests statement

YCC, YFeC, HWC, RSW, QWT, and FYC are employees of Oneness Biotech. CFY, CCY, YLC, and TYL are employees of Microbio (Shanghai) Biotech. The other authors declare that they have no conflict of interest.

## For more information

http://www.ibms.sinica.edu.tw/pan-chyr-yang/
https://www.gisaid.org/
https://www.ecdc.europa.eu/en/covid-19/variants-concern
https://www.ncbi.nlm.nih.gov/sars-cov-2/

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
