## [Review Process File · EMBO Molecular Medicine]

A siRNA targets and inhibits a broad range of SARS-CoV-2 infections including Delta variant

Pan-Chyr Yang, Yi-Chung Chang, Chi-Fan Yang, Yi-Fen Chen, Chia-Chun Yang, Yuan-Lin Chou, Hung-Wen Chou, Tein-Yao Chang, Tai-Ling Chao, Shu-Chen Hsu, Siman leong, Ya-Min Tsai, Ping-Cheng Liu, Yuan-Fan Chin, Jun-Tung Fang, Han-Chieh Kao, Hsuan-Ying Lu, Jia-Yu Chang, Ren-Shiuan Weng, Qian-Wen Tu, Fang-Yu Chang, Kuo-Yen Huang, Tong-Young Lee, and Sui-Yuan Chang

DOI: 10.15252/emmm.202115298

Corresponding authors: Pan-Chyr Yang (pcyang@ntu.edu.tw) , Sui-Yuan Chang (sychang@ntu.edu.tw)

Review Timeline:

Submission Date:	18th Oct 21
Editorial Decision:	12th Nov 21
Revision Received:	27th Dec 21
Editorial Decision:	28th Jan 22
Revision Received:	1st Feb 22
Accepted:	4th Feb 22

Editor: Zeljko Durdevic

Transaction Report:

12th Nov 2021

Dear Prof. Yang,

Thank you for the submission of your manuscript to EMBO Molecular Medicine. We have now received feedback from the three reviewers who agreed to evaluate your manuscript. As you will see from the reports below, the referees acknowledge the interest of the study but also raise important critique that should be addressed in a major revision.

We would welcome the submission of a revised version within three months for further consideration. Please let us know if you require longer to complete the revision.

Please use this link to login to the manuscript system and submit your revision: Link Not Available

I look forward to receiving your revised manuscript.

Yours sincerely,

Zeljko Durdevic

**** Reviewer's comments ****

Referee #1 (Comments on Novelty/Model System for Author):

The models used are standard including the Veri cell culture systems and the hACE2-K18 mouse model.

Referee #1 (Remarks for Author):

Overall, this is a very nice piece of work that has impressive results in the inhaled use of siRNAs for SARS-CoV-2. The work is well done and covers all the bases in terms of results I would expect to see. The discussion however needs a complete rewrite and I have some suggestions and comments below that should be addressed.

IC50s - it is stated "Half-maximal inhibitory concentration (IC50) for C6G25S, C8G25S and C10G31A were determined as 0.17, 1.03 and 0.76 nM, respectively (Fig 1D)." (p5 line 5) - please correct this to note that this refers only to envelope gene expression and not viral production. In terms of IC50 I would actually prefer to see these measures presented as reductions in viral replication as the qPCR for E gene is a surrogate marker for virus output and therefore one step removed from this. Plaque reductions have already been performed and this would be a more robust marker of actual viral suppression. The authors actually demonstrate in Fig 4 where virus copies don't match virus output. This should improve the IC50 rates if this is consistent with the in vivo work.

P6 line 10 - while direct delivery is not a problem the stated logic for using this approach is that particle-based delivery will induce immune responses. This isn't correct as naked siRNAs can induce immune responses as well. I would perhaps rephrase to say since PBMCs didn't show any inflammatory responses (Fig S3) and others have delivered siRNAs to the lung naked before (Bitko et al) it was decided to use this approach rather than a more complex particle-based delivery system. Animal experiments. Very nice data. However, to call this post-exposure is a bit of a stretch given treatment is immediately after infection. Stanley Perlman, in their primate studies for SARS-CoV-1, called this approach "co-treatment". However, I acknowledge that there is no set definition for post-exposure but I would expect that it would be at least some hours after infection if not 24 hours.

Fig 4 - please add tick markers between the log scale indicators

Would it be useful to show that mice lungs treated with siRNAs only along had no induction of Interferon alpha or other antiviral cytokines just to exclude the possibility of an indirect antiviral effect? Fig S3 does this with PBMCs but of course these are not the cells being treated. Much of the flu siRNA literature was beset with this issue. Perhaps mentioning this issue in the discussion would be adequate.

Discussion. I found this to be entirely unsatisfactory. They should contextualise the work with the previous literature. They should refer to the previous SARS-COV-1 siRNA studies and the recently published works on SARS-CoV-2 siRNA (a quick search found at least 2 previously papers). This work is sufficiently different from these to be novel and the authors should outline this. . Please don't just restate results nor introduce new results (e.g. Fig S6 and 7)

Referee #2 (Comments on Novelty/Model System for Author):

The impact of the work is high if delivery is improved. The humanized mouse model is one of the best models available at the moment, but the novelty of delivering naked siRNA to the lung is low as it was described for many other respiratory virus infections previously, and has failed in clinical trials. Therefore, the delivery needs to be improved to potentially meet clinical

endpoints later on.

Referee #2 (Remarks for Author):

The manuscript "Title: A C6G25S siRNA targets and inhibits a broad range of SARS CoV 2 infections including Delta variant" by Chang et al describes a topical research problem addressing a clear medical need. The authors identified siRNA sequences against a conserved region of SARS1/2 to develop an inhalable siRNA-based antiviral therapy. The strong point of the manuscript are the efficacy studies, however, several virologic and RNA delivery aspects are not conclusively addressed in the manuscript and need to be improved before the manuscript can be published.

Major comments:

The authors nebulized siRNA solutions with an Aeroneb but neither investigated how nebulization impacted siRNA integrity nor aerodynamic properties of the aerosol. It is suggested that the siRNA yield/recovery after nebulization as well as integrity and MMAD, GSD and fine particle fraction of the aerosol be investigated understand which dose of intact siRNA was even administered to the animals.

The authors treated the animals with 2-3 doses of 50 µg each, which compares to a dose of 50 µg per 20 g body weight, equal to 2.5 mg siRNA per kg body weight, and 175 mg daily RNA dose for a 70 kg adult. This seems to be outside the range of what any healthcare system could shoulder. Could the authors provide in vivo results with a more efficient delivery approach?

Page 7, line 5 onwards and Figure 3: The materials and methods part mentions 5 animals per group, but Figures 4E and F only reflect an n of 3. Based on the siRNA amount recovered from the lungs, the average seems comparable for intranasal vs. inhalation delivery, which is not reflected by Figures 3A and 3B. Are the authors certain that the quantification method via stem-loop PCR results in reliable data? Did they perform internal standard experiments with tissues spiked with known amounts of siRNA?

Lung injury was qualitatively investigated by microscopic assessment of lung tissue sections. Rather than a qualitative assessment, a quantitative analysis of cells and cytokines in the lung lavage fluid is suggested.

It is not clear why different species (mice, rats and human PMBCs) were used for the same purpose of investigating toxicity and why lung lavage, a very standard technique for quantitative results on toxicity, was not performed in any of the animals.

1. On page 4, line 11ff the authors write:

Further evaluating the location of the siRNA binding sites on the vital genes involved in virus replication and infection, 374 located in regions encoding the viral leader, papain-like protease, 3C-like protease, RNA dependent RNA polymerase (RdRp), helicase, spike protein, and the envelope protein were isolated

Due to the specific (discontinuous) replication of coronaviruses, all viral transcripts carry a common 5' (Leader Sequence) and 3' end (N/ORF10 and 3'UTR) which was also confirmed for SARS-CoV-2 (e.g. <https://doi.org/10.1016/j.cell.2020.04.011>). As the siRNA targets the viral RNAs and not proteins, it is not possible to specifically target the indicated genes. Thus, the described procedure to me makes no sense.

The authors in many instances appear to overstate the role of escape mutations:

Two examples:

a) E.g. in the last paragraph in the discussion they write "vaccine-resistant variants continue to..." After all the most widely used vaccines strongly reduce the mortality and hospitalization also by the Delta variant. Also, even in case that new and more relevant escape variants should occur, most of the vaccines could also be quickly adapted (e.g. mRNA vaccines). The authors should thus down tone the statements on the limitations of the vaccines and the role of the delta variant.

b) On page 3 line 6ff the authors write: According to the United States Centers for Disease Control and Prevention, 90% of 469 new infections in Barnstable County, Massachusetts were caused by the Delta variant, and among those, 74% were already fully vaccinated (Brown, Vostok et al., 2021).

I strongly believe that such a presentation is misleading. The data could only be fully interpreted if one would know the fraction of vaccinated people.

On page 3 line 8ff they write: Furthermore, vaccinated and unvaccinated people carry similar viral loads. These data suggest that the current vaccine strategy fails to halt SARS-CoV-2 transmission. What do the authors want to achieve with their siRNA therapy? Which is the patient group they are planning to use the drug for? Halt transmission in otherwise asymptomatic people or people with mild symptoms?

What is the goal of the transcriptome analysis? While I understand that it is of interest to identify potential off-target genes, I

believe it is very hard to say that a low number of hits in their analysis corresponds to low off-target activity. E.g. it appears to me that the authors have set the threshold to define a significant de-regulated gene very high (threefold difference, FDR 0.001). Also, I cannot find any information on the number of biological replicates they analyzed, which would also influence the number of genes which one would find de-regulated. Thus, while I see the purpose of this analysis to define potential off-targets, I do not see that a low number of significant hits (by their definition) few off-target activity.

There is in general very few information on the use of controls.

a) In their cell culture experiment shown in Figure 1D,E,F and Figure 2B: Were the values set in relation to a control siRNA?
b) Also, I cannot find any information in the figure legend of Figures describing the in vivo data: E.g. in Figure 4 and Figure 5F,H what the control consisted of. Is this a non-relevant or scrambled siRNA? Or an untreated animal? This would be an important information to judge if the therapy was specific or if it could have been a result of an unspecific immunostimulatory effect.

Minor comments:

Page 3 lines 3/4

(University, 2021), please check if the citation is displayed in a correct way

Referee #3 (Comments on Novelty/Model System for Author):

The development of new therapeutics against SARS-CoV-2 variants is an important topic. Authors developed an inhalable siRNA that inhibits SARS-CoV-2 VOCs at picomolar ranges in vitro and is active in ACE2-mice. Experiments were done with infectious virus under BSL3 conditions, the siRNA is probably the first to be shown to be active in vivo against CoV2, without causing severe side effects.

Referee #3 (Remarks for Author):

Major Issues:

1. Authors claim to have performed a "post-exposure treatment" but actually it looks as if the drug and the virus were administered simultaneously. This must be clarified throughout the manuscript. Post exposure implies that virus has already established infection, and this seems not to be the case.
2. What are the disadvantages or limitations of RNAs as inhalative drugs? Are there similar approaches against other (respiratory) viruses?

Minor Issues:

Title: Remove "C6G25S"

abstract emergence of SARS-2 (add-2)

page 3 line 10: "fails to" is too strong statement

page 3 line 11: vaccines are not therapeutics

page 4 line 20. Why "E" selected for amplification?

page 5 line 12: what is "true" meaning?

page 5 line 13: rephrase, C6 does not inhibit the E gene but amplification of viral RNA using primer binding to E?

page 5 line 15, siRNA with potent antiviral activity

page 6 line 2: "front", better 5 prime?

page 14 line 33: which SARS-CoV-2 isolate/strain/VOC

page 14: 26ff: authors must explain how exactly the animal experiments were performed in the "post exposure" setting.

Point-by-point response to the Reviewers' comments

(Submission ID: EMM-2021-15298)

Response for Reviewer 1:

Overall, this is a very nice piece of work that has impressive results in the inhaled use of siRNAs for SARS-CoV-2. The work is well done and covers all the bases in terms of results I would expect to see. The discussion however needs a complete rewrite and I have some suggestions and comments below that should be addressed.

We are very thankful to the **Reviewer 1** for the positive comments that help us to strengthen the discussion of our manuscript. We have carefully replied all the comments and rewrite the discussion section. The point-by-point responses are provided directly afterward.

Comment 1:

IC50s - it is stated "Half-maximal inhibitory concentration (IC50) for C6G25S, C8G25S and C10G31A were determined as 0.17, 1.03 and 0.76 nM, respectively (Fig 1D)." (p5 line 5) - please correct this to note that this refers only to envelope gene expression and not viral production. In terms of IC50 I would actually prefer to see these measures presented as reductions in viral replication as the qPCR for E gene is a surrogate marker for virus output and therefore one step removed from this. Plaque reductions have already been performed and this would be a more robust marker of actual viral suppression. The authors actually demonstrate in Fig 4 where virus copies don't match virus output. This should improve the IC50 rates if this is consistent with the *in vivo* work.

Response

We thank the reviewer for pointing out that plaque reductions would be a more robust marker of viral suppression. We have modified Fig 1D and Fig 2B as advised to present IC50 via plaque reduction assay.

Original statement in p5 line 5:

Half-maximal inhibitory concentration (IC50) for C6G25S, C8G25S and C10G31A were determined as 0.17, 1.03 and 0.76 nM, respectively (Fig 1D).

Revised the statement in page 6 line 10:

Half-maximal inhibitory concentrations (IC₅₀) for C6G25S, C8G25S and C10G31A, determined by plaque reduction assay were 0.07, 0.24 and 0.12 nM, respectively (Fig 1D).

Comment 2:

P6 line 10 - while direct delivery is not a problem the stated logic for using this approach is that particle-based delivery will induce immune responses. This isn't correct as naked siRNAs can induce immune responses as well. **I would perhaps rephrase to say since PBMCs didn't show any inflammatory responses (Fig S3) and others have delivered siRNAs to the lung naked before (Bitko et al) it was decided to use this approach rather than a more complex particle-based delivery system.**

Response

We appreciate the reviewer's valuable suggestion. We have modified the sentences as shown below:

Original statement in P6 line 10:

Direct delivery of C6G25S via intranasal instillation (IN) or aerosol inhalation (AI) were next implemented as considering that the siRNA carriers, such as virus-like particles, lipid nanoparticles and cell penetrating peptides, could cause adverse immune stimulation or cytotoxicity.

Revised the statement in page 7 line 18:

As peripheral blood mononuclear cells (PBMCs) did not show any significant inflammatory responses (Appendix Fig S3) to treatment with C6G25S, and because others have successfully delivered naked siRNA to the lung via intranasal instillation (IN) or aerosol inhalation (AI), we decided to use a naked siRNA approach rather than a more complex delivery system, such as virus-like particles, lipid nanoparticles or cell penetrating peptides that have been reported to cause adverse immune stimulation or cytotoxicity (Farkhani, Shirani et al., 2016, Slutter, Bal et al., 2011, Vangasseri, Cui et al., 2006, Wilson, 2009).

Reference

- Farkhani SM, Shirani A, Mohammadi S, Zakeri-Milani P, Shahbazi Mojarrad J, Valizadeh H (2016) Effect of poly-glutamate on uptake efficiency and cytotoxicity of cell penetrating peptides. *IET Nanobiotechnol* 10: 87-95
- Slutter B, Bal SM, Ding Z, Jiskoot W, Bouwstra JA (2011) Adjuvant effect of cationic liposomes and CpG depends on administration route. *J Control Release* 154: 123-30
- Vangasseri DP, Cui Z, Chen W, Hokey DA, Falo LD, Jr., Huang L (2006) Immunostimulation of dendritic cells by cationic liposomes. *Mol Membr Biol* 23: 385-95
- Wilson JM (2009) Lessons learned from the gene therapy trial for ornithine transcarbamylase deficiency. *Mol Genet Metab* 96: 151-7

Comment 3:

Animal experiments. Very nice data. However, to call this post-exposure is a bit of a stretch given treatment is immediately after infection. Stanley Perlman, in their primate studies for SARS-CoV-1, called this approach "co-treatment". However, I acknowledge that there is no set definition for post-exposure but I would expect that it would be at least some hours after infection if not 24 hours.

Response

We thank the reviewer for pointing out that it might be inappropriate to use postexposure treatment in our case.

Details of our post-exposure treatment are presented below:

Mice were first anesthetized and infected intranasally with SARS-CoV-2. After 30 mins of recovery, mice were placed in the chamber to undergo 30 min of inhalation treatment.

Given that the interval between siRNA delivery and virus infection was short, we have replaced the “post-exposure treatment” with “co-treatment” as advised.

We have modified all the descriptions of postexposure treatment to co-treatment. In total, there are 14 sites identified in the Main text and 0 in the Appendix, as listed below:

Site	Original	Revised
P1 Line 8	post-exposure treatment in K18-hACE2-transgenic mice	co-treatment in K18-hACE2-transgenic mice
P7 Line 21	post-exposure treatment in vivo.	co-treatment in vivo.
P8 Line 2	postexposure administration of C6G25S	co-treatment of C6G25S
P8 Line 5	96.2% in the postexposure group	96.2% in the co-treatment group
P8 Line 7	96% was observed in the postexposure group	96% was observed in the co-treatment group
P8 Line 13	Two post-exposure groups	Two co-treatment groups
P8 Line 19	prophylactic and post-exposure treatment	prophylactic and co-treatment
P11 Line 16	In the post-exposure treatment	In the co-treatment
P13 Line 46	for postexposure treatment.	for co-treatment.
P14 Line 34	For postexposure treatment	For co-treatment
P18 Line 19	post-exposure treatment in hACE2 transgenic mice.	co-treatment in hACE2 transgenic mice.
P23 Line 11	Fig. 4. Prophylactic and post-exposure	Fig. 4. Prophylactic and co-treatment
P23 Line 18	postexposure treated with	co-treatment with
P23 Line 23	Postexposure treatment of C6G25S	Co-treatment of C6G25S

Revised statement in Material and Methods section (page 18 line 22)

For co-treatment, mice were first anesthetized with Zoletil/Dexdomitor and infected intranasally with 10^4 PFU of SARS-CoV-2. After 30 min of recovery, mice were placed in the chamber to perform 30 min of inhalation treatment (D0). Mice were treated with aerosolized siRNA at D0 and 1-day post-infection. Infected mice were sacrificed to collect their lungs at two days post-infection.

Comment 4:

Would it be useful to show that mice lungs treated with siRNAs only along had no induction of Interferon alpha or other antiviral cytokines just to exclude the possibility of an indirect antiviral effect? Fig S3 does this with PBMCs but of course these are not the cells being treated. Much of the flu siRNA literature was beset with this issue. Perhaps mentioning this issue in the discussion would be adequate.

Response

We appreciate the reviewer's valuable suggestion. Two additional experiments have been conducted to further characterize the immunogenicity of C6G25S.

Study 1

This study showed that the efficacy dose of C6G25S does not stimulate IL-6, TNF- α , IFN- α and IFN- γ expression in ICR mice. In contrast, poly(I:C) and unmodified siRNA are capable of triggering a cytokine response. The data are presented in **Fig EV3**.

Study 2

This study showed that no significant immune cells infiltration in bronchoalveolar lavage or cytokine induction in lung tissue was observed in ICR mice challenged with up to 75 mg/kg of C6G25S. The data are presented in **Fig EV4**.

New information has been added to four different sections in the revised manuscript:

(1) Figure EV3A (2) Figure EV4 (3) Result section (4) Materials and Methods section

(1) **Figure EV3A**

Figure EV3A. C6G25S does not activate non-specific immune response against SARS-CoV-2.

ICR mice (n = 3) were treated with vehicle alone (saline), 2.5 mg/kg of poly(I:C) and negative control siRNA (an unmodified siRNA sequence used in the cell-based virus inhibition assay) via high volume (50 μ L) of intranasal administration. Lung tissue was collected at 48 hours after treatment. The mRNA expression of IL-6, TNF- α , IFN- α and IFN- γ was quantified by RT-PCR. Data are presented as mean \pm SD. *P* value by Student *t* test.

(2) **Figure EV4**

Figure EV4. No significant immune stimulation was observed in mice treated with high-dose of C6G25S.

- A. ICR mice (n = 3) treated with saline control or different dosages (20, 40, 75 mg/kg) of C6G25S by intranasal instillation. The bronchoalveolar lavage (BAL) was collected at day 2 after treatment. Different WBC cell counts were analyzed by a hematology analyzer. The cell counts are expressed as mean \pm SD. All of the different treatment groups are statistically insignificant compared with the control.
- B. Lung tissues of each mouse were collected, and RNA was extracted. The mRNA expression of IL-6, TNF- α , IFN- α and IFN- γ was quantified by RT-PCR. The relative amount of each cytokine is presented as mean \pm SD. All of the different treatment groups are statistically insignificant compared with the control.

(3) **Result section** (page 12 line 3)

To investigate the potential local immune response of C6G25S, lungs from ICR mice treated with efficacy dose of C6G25S were analyzed for the expression of pro-inflammatory cytokines including IL-6, TNF- α , IFN- α and IFN- γ . C6G25S treatment did not induce any cytokine expression. In contrast, *Poly(I:C)* induced the expression of IL-6, TNF- α and IFN- γ and an extensively used control siRNA (Kermorgant, Zicha et al., 2004) was shown to induce 8-fold of IL-6 expression (**Fig EV3A**). Despite the induction of IL-6, control siRNA did not show any inhibition of the viral RNA amplification that might be evoked through nonspecific immune responses (**Figure EV3B**). Furthermore, administration up to 75 mg/kg of C6G25S did not induce immune cell infiltration in bronchoalveolar lavage fluid (**Fig EV4A**) and pro-inflammatory cytokines expression in lung tissue (**Fig EV4B**).

(4) Materials and Methods section

Quantification of cytokine mRNA via RT-qPCR (page 19 line 43)

The lungs were homogenized in RLT buffer (Qiagen) using a TissueLyser II (Qiagen) at 4°C and clarified by centrifugation at 15,000 g for 15 min at 4 °C. Total RNA was extracted with a RNeasy Micro Kit (Qiagen) following the manufacturer's instructions. Reverse transcription reaction was conducted with a Maxima First-Strand cDNA Synthesis kit (Thermo Fisher Scientific) and 1 ug of total cellular RNA. qPCR was carried out on a LightCycler 480 using SYBR Green I Master (Roche Diagnostics) with 1:5 dilutions of cDNA. Primer sets used to detect cytokine genes are shown in **Appendix Table S7**. Each sample was assayed in triplicate to determine an average threshold cycle (Ct) value. Gene expression fold change was calculated using the $\Delta\Delta C_t$ method. The mRNA level of each gene was normalized to constitutively expressed GAPDH mRNA.

In vitro and in vivo evaluation of immunogenicity of C6G25S (page 20 line 18)

To investigate if C6G25S may cause acute local immune response in the lungs, 7-week-old male Blt \times :CD1(ICR) mice obtained from Shanghai Model Organisms Center (Shanghai, China) were intranasal instilled with vehicle alone (saline), poly IC at 2.5 mg/kg (positive control) or C6G25S at 0, 20, 40, or 75 mg/kg. After 48 h, mice were euthanized by isoflurane inhalation and bronchoalveolar lavage fluid (BALF) collected via intratracheal infusion of 0.8 mL of PBS. BALF was obtained by retracting the piston of the syringe three times and centrifuged at 450 g for 5 min at 4 °C. The resulting cell pellet was resuspended in 200 μ L of PBS and analyzed immediately using an automated hemacytometer (BX3010, Sysmex). The supernatant was stored at -80 °C. The mRNA levels of cytokines in lungs were determined via RT-qPCR. NC siRNA (Kermorgant et al., 2004) and NC siRNA2 were used as controls. NC siRNA2 was designed by blasting the database for no match to human sequences and SARS-CoV-2 genome. The sense and antisense sequence of NC siRNA was 5'-UUC UCC GAA CGU GUC ACG UTT-3' and 5'-ACG UGA CAC GUU CGG AGA ATT-3'. NC siRNA2 was 5'-UUC GAC CGG UAU AUG GUA GTT-3' and 5'-CUA CCA UAU ACC GGU CGA ATT-3'.

Appendix Table S7.

Appendix Table S7. RT-qPCR primers for the quantification of cytokine mRNA

Genes	Forward primer (5'-3')	Reverse primer (5'-3')
IL6	TACCACTTCACAAGTCGGAGGC	CTGCAAGTGCATCATCGTTGTTC
TNF α	GGTGCCTATGTCTCAGCCTCTT	GCCATAGAAGTGTGATGAGAGGGGAG
IFN γ	CGGCACAGTCATTGAAAGCCTA	GTTGCTGATGGCCTGATTGTC
IFN α	GGATGTGACCTTCCTCAGACTC	ACCTTCTCCTGCGGGAATCCAA
GAPDH	CGACTTCAACAGCAACTCCCACTCTTCC	TGGGTGGTCCAGGGTTTCTTACTCCTT

Reference

Kermorgant S, Zicha D, Parker PJ (2004) PKC controls HGF-dependent c-Met traffic, signalling and cell migration. *EMBO J* 23: 3721-34

Comment 5:

Discussion. I found this to be entirely unsatisfactory. They should contextualise the work with the previous literature. They should **refer to the previous SARS-COV-1 siRNA studies and the recently published works on SARS-CoV-2 siRNA (a quick search found at least 2 previously papers)**. This work is sufficiently different from these to be novel and the authors should outline this. Please don't just restate results nor introduce new results (e.g. Fig S6 and 7)

Response

We appreciate the reviewer's valuable suggestion. We have raised the related discussion about previous siRNA works in page 13 (Discussion section).

New statement in Discussion section (page 13)

IN delivery of naked siRNA has been proven to be an effective approach to prevent and treat SARS-CoV-1 infection in non-human primates (Rhesus macaque) (Li, Tang et al., 2005). Because the SARS-CoV-1 outbreak had been brought under control in a short period of time, the therapeutics has not proceeded to clinical use. Recently, several studies testing siRNA against SARS-CoV-2 infection have been published (Niktab, Haghparast et al., 2021, Shawan, Sharma et al., 2021, Tolksdorf, Nie et al., 2021, Wu & Luo, 2021). While most researchers have pursued a siRNA design that has been validated through *in vitro* cell-based experiments, only two publications have assessed the efficacy of their siRNA in an animal model. One publication used intravenous administration of LNPs-siRNA to inhibit SARS-CoV-2 infection in hACE2 transgenic mice (Idris, Davis et al., 2021), and the viral titer was reduced by about a log of magnitude at day 3 post-infection by prophylactic treatment. The other study used positive-charged dendrimer to carry siRNA and treated SARS-CoV-2-infected Syrian hamster via inhalation (Khaitov, Nikonova et al., 2021). The viral titer was reduced by 30% at day 2 post-infection. The potency of viral inhibition reported in these studies was not as significant as ours. Moreover, LNPs and positive charge dendrimer have been found to be capable of inducing immune or cell toxicity (Kedmi, Ben-Arie et al., 2010, Kharwade, Badole et al., 2021), which might limit the safety window for dosing. These properties also increase the difficulty and cost of industrial production. On the contrary, the feasibility and safety of respiratory-delivered, unmodified naked siRNA has been demonstrated in animal models and in clinical trials (DeVincenzo, Lambkin-Williams et al., 2010, Li et al., 2005). Taken together,

fully modified C6G25S with low immunogenicity (**Figure EV3A and EV4**) and reduced off-target effect (**Appendix Fig S1**) was developed for naked delivery to the respiratory system as a safe, effective and feasible approach against SARS-CoV-2.

Reference

DeVincenzo J, Lambkin-Williams R, Wilkinson T, Cehelsky J, Nochur S, Walsh E, Meyers R, Gollob J, Vaishnav A (2010) A randomized, double-blind, placebo-controlled study of an RNAi-based therapy directed against respiratory syncytial virus. *Proc Natl Acad Sci U S A* 107: 8800-5

Idris A, Davis A, Supramaniam A, Acharya D, Kelly G, Tayyar Y, West N, Zhang P, McMillan CLD, Soemardy C, Ray R, O'Meally D, Scott TA, McMillan NAJ, Morris KV (2021) A SARS-CoV-2 targeted siRNA-nanoparticle therapy for COVID-19. *Mol Ther* 29: 2219-2226

Kedmi R, Ben-Arie N, Peer D (2010) The systemic toxicity of positively charged lipid nanoparticles and the role of Toll-like receptor 4 in immune activation. *Biomaterials* 31: 6867-75

Khaitov M, Nikonova A, Shilovskiy I, Kozhikhova K, Kofiadi I, Vishnyakova L, Nikolskii A, Gattinger P, Kovchina V, Barvinskaia E, Yumashev K, Smirnov V, Maerle A, Kozlov I, Shatilov A, Timofeeva A, Andreev S, Koloskova O, Kuznetsova N, Vasina D et al. (2021) Silencing of SARS-CoV-2 with modified siRNA-peptide dendrimer formulation. *Allergy* 76: 2840-2854

Li BJ, Tang Q, Cheng D, Qin C, Xie FY, Wei Q, Xu J, Liu Y, Zheng BJ, Woodle MC, Zhong N, Lu PY (2005) Using siRNA in prophylactic and therapeutic regimens against SARS coronavirus in Rhesus macaque. *Nat Med* 11: 944-51

Meng Z, Lu M (2017) RNA Interference-Induced Innate Immunity, Off-Target Effect, or Immune Adjuvant? *Front Immunol* 8: 331

Niktab I, Haghparast M, Beigi MH, Megraw TL, Kiani A, Ghaedi K (2021) Design of advanced siRNA therapeutics for the treatment of COVID-19. *Meta Gene* 29: 100910

Robbins M, Judge A, Liang L, McClintock K, Yaworski E, MacLachlan I (2007) 2'-O-methyl-modified RNAs act as TLR7 antagonists. *Mol Ther* 15: 1663-9

Shawan M, Sharma AR, Bhattacharya M, Mallik B, Akhter F, Shakil MS, Hossain MM, Banik S, Lee SS, Hasan MA, Chakraborty C (2021) Designing an effective therapeutic siRNA to silence RdRp gene of SARS-CoV-2. *Infect Genet Evol* 93: 104951

Tolksdorf B, Nie C, Niemeyer D, Rohrs V, Berg J, Lauster D, Adler JM, Haag R, Trimpert J, Kaufer B, Drosten C, Kurreck J (2021) Inhibition of SARS-CoV-2 Replication by a Small Interfering RNA Targeting the Leader Sequence. *Viruses* 13

Wu R, Luo KQ (2021) Developing effective siRNAs to reduce the expression of key viral genes of COVID-19. *Int J Biol Sci* 17: 1521-1529

Comment 6:

Fig 4 - please add tick markers between the log scale indicators

Response

We thank for the reviewer's suggestion. We have added tick markers between the log scale indicators in all subfigures in Fig 4 as shown below

Response for Reviewer 2:

The impact of the work is high if delivery is improved. The humanized mouse model is one of the best models available at the moment, but the novelty of delivering naked siRNA to the lung is low as it was described for many other respiratory virus infections previously, and has failed in clinical trials. Therefore, the delivery needs to be improved to potentially meet clinical endpoints later on.

We are very thankful to the Reviewer 2 for the insightful comments to make our work more complete. The reason we use naked delivery of fully modified C6G25S as our first generation medicine was described in the response section of comment 2. In the meantime, we have carefully replied all the comments and point-by-point responses are provided directly afterward.

Comment 1:

The authors nebulized siRNA solutions with an Aeroneb but neither investigated how nebulization impacted siRNA integrity nor aerodynamic properties of the aerosol. It is suggested that the siRNA yield/recovery after nebulization as well as integrity and MMAD, GSD and fine particle fraction of the aerosol be investigated understand which dose of intact siRNA was even administered to the animals.

Response

We thank for the reviewer's valuable suggestion. We had conducted extra studies to further characterize the aerosolized C6G25S, including the C6G25S concentration, integrity, efficacy, median mass aerodynamic diameter (MMAD), geometric standard deviation (GSD), and fine particle fraction (FPF) after nebulization.

New information has been added to three different sections in the revised manuscript:

(1) Figure EV1 (2) Result section (3) Materials and Methods section

(1) Figure EV1:

Figure EV1. Characterization and effectiveness of C6G25S in inhalation aerosol.

- A. The concentration of C6G25S before and after nebulization. The siRNA aerosol generated from 1 mL saline containing 125, 25, 5, 1 and 0.2 mg/mL of C6G25S was collected. The concentration of C6G25S before and after nebulization was measure by OD260.
- B. The integrity of siRNA before and after nebulization was analyzed via HPLC. The siRNA aerosol generated from 1 mL saline containing 6 mg/mL C6G25S was collected and analyzed compared with that before nebulization. Neither denaturation nor degradation was found after nebulization. DS control: Stock of double strand C6G25S. SS control: Antisense strand of C6G25S. N(-5) control: Double strand C6G25S with 5 bases truncated in both strands.
- C. The effectiveness of C6G25S before and after nebulization was determined via the inhibition of viral RNA. The siRNA aerosol was generated from 0.5 mL saline containing C6G25S 0.5 mg/mL. The RNA concentration was then determined by OD260 before nebulization and after siRNA aerosol was collected. Vero E6 cells were transfected with C6G25S (before or after nebulization) at different concentration for 24 h before infection with SARS-CoV-2 at a multiplicity of infection (MOI) of 0.1. The control cells were transfected with negative siRNA. The expression of viral envelope gene in the virus-infected cells was quantified by RT-qPCR 24 h after infection.
- D. Particle size distribution of nebulized siRNA aerosol generated from Aeroneb nebulizer and 2 mL normal saline containing 6 mg/mL C6G25S siRNA was analyzed using the next generation impactor (NGI). The experiment was performed in a laboratory accredited under ISO17025.

(2) Result section (page 8 line16)

After C6G25S was nebulized, we collected the condensed aerosol and measured C6G25S by OD260. The concentration of C6G25S was the same before and after nebulization (Fig EV1A). The integrity of the siRNA, detected via HPLC, was not affected by nebulization (Fig EV1B). The effectiveness of the siRNA after nebulization was the same as that before nebulization, evaluated by the inhibition of viral envelope gene expression in Vero E6 cells (Fig EV1C). Particle size distribution of the siRNA aerosol is presented in Fig EV1D. The particle size of the siRNA aerosol generated by the nebulizer had a mass median aerodynamic diameter (MMAD) of 4.725 μm , a geometric standard deviation (GSD) of 2.376 μm with a fine particle fraction (FPF; $<5 \mu\text{m}$) of 51.94%. The drug recovery rate was 90%. The nebulization rate was maintained at 0.5 mL/min when the concentration of C6G25S was ≤ 30 mg/mL and reduced significantly at higher concentrations (Appendix Fig S2A).

(3) Materials and Methods section

In-vitro aerodynamic deposition study (page 18 line 7)

The *in vitro* aerodynamic attributes including mass median aerodynamic diameter (MMAD), geometric standard deviation (GSD) and fine particle dose, fine particle fraction (FPF) were measured at MicroBase Technology (Taoyuan City, Taiwan) using the next generation impactor (NGI) and a USP induction port (Copley Scientific, Nottingham, UK). The NGI was assembled and operated in accordance with USP General Chapter 1601 to assess the drug delivered.

HPLC conditions for the determination of siRNA (page 19 line 43)

The determination of siRNA was performed with a Waters ACQUITY Arc HPLC system equipped with a PDA detector. Chromatographic separation was achieved on Shodex KW-802.5 (8.0 mm I.D. x 300 mm) column and potassium phosphate buffer (3 mM sodium phosphate, 150 mM sodium chloride, 1.05 mM potassium phosphate, pH 7.4) as a mobile phase. The HPLC conditions included a column temperature at 40 °C, flow rate of 1.0 mL/min, UV detection at 260 nm, injection volume of 10 μL , and run time of 15 min. C6G25S sense strand (n, full length), C6G25S antisense strand (n, full length), C6G25S sense strand with 5 bases truncated (n-5), C6G25S antisense strand with 5 bases truncated (n-5), C6G25S double strand (n, full length) and C6G25S double strand with 5 bases truncated in each strand (n-5, fully complementary) were used as controls.

Comment 2:

The authors treated the animals with 2-3 doses of 50 µg each, which compares to a dose of 50 µg per 20 g body weight, equal to 2.5 mg siRNA per kg body weight, and 175 mg daily RNA dose for a 70 kg adult. This seems to be outside the range of what any healthcare system could shoulder. Could the authors provide *in vivo* results with a more efficient delivery approach?

Response

We thank the reviewer for the valuable consideration.

The indicated dose may be achievable, and effective delivery systems are undergoing investigation. The human dosage will be further carefully identified in clinical trials.

A similar dose had been utilized in clinical trials of Alnylam's siRNA (ALN-RSV01). ALN-RSV01 had been administrated intranasally up to 150 mg daily in phase II study (DeVincenzo et al., 2010) and tested up to 3 mg/kg of daily dose for aerosolized inhalation (https://osp.od.nih.gov/wp-content/uploads/2014/01/2_Gollob.pdf; page 17). Moreover, C6G25S solution can be prepared up to 30 mg/mL without affecting the nebulization speed (0.5ml/min) (Appendix Fig S2A) and it suggest that the system can deliver an even higher dose than the indicated 175 mg if needed.

Appendix Fig S2A

Different concentrations of C6G25S solutions, including 150, 50, 40, 30, 6, 1.2, 0.24 and 0.048 mg/mL in normal saline, were analyzed for nebulization rate in a mesh nebulizer.

In addition, it might be inappropriate to predict human dose directly by the ratio of body mass between human and mice. Allometric scaling has been used widely as the basis for extrapolation of drug dosage that might be expected to produce the equivalent biological effects (Phillips, 2017). A review applied allometric scaling for human dose projections from pre-clinical data of compounds that are delivered by inhalation (Phillips, 2017). It used the equation of $(X_h = X_a(M_h/M_a)^{1-b})$ to predict human dose. X_h is the human drug dose normalized to body mass ($\mu\text{g}/\text{kg}$). M_h is the human body mass (kg). X_a is the animal drug dose per unit body mass ($\mu\text{g}/\text{kg}$). M_a is the animal body mass (kg), and b is the fixed allometric exponent, which is 0.67. The

efficacious delivered dose in our study is 2.5 mg/kg in mice. Based on this allometric scaling calculation, the projected equivalent dose will be 0.17 mg/kg in human. However, the effective dose in animal studies is only a reference, and the human dosage should be further carefully identified in clinical trials.

Naked siRNA delivery and more efficient delivery approach

ALN-RSV01 is the only example that uses naked siRNA for pulmonary delivery in clinical trials. Although the drug marginally missed ($P = 0.058$) the primary endpoint in reducing the occurrence of bronchiolitis obliterans syndrome (BOS) in RSV-infected lung transplant recipients at 180 days. However, the data of 90 days in the same trial was significant ($P = 0.027$) (Gottlieb, Zamora et al., 2016). Moreover, the risk of RSV infection was significantly ($P = 0.0069$) reduced by ALN-RSV01 administration in another phase 2 trial (DeVincenzo et al., 2010). These data suggest that naked siRNA delivery had been proved to be a safe approach and could achieve clinically meaningful improvement.

Furthermore, ALN-RSV01 was an unmodified siRNA. Unmodified siRNA was known to be nuclease sensitive and immunogenic. The immunogenicity of ALN-RSV01 had limited its final dose at 0.6 mg/kg in a phase 2b study (https://osp.od.nih.gov/wp-content/uploads/2014/01/2_Gollob.pdf ; page 18). On the contrary, C6G25S is a fully-modified siRNA that is nuclease resistant and low immunogenic, thus might allow us to apply a higher dose to improve the clinical outcome.

However, we agree with the reviewer that if a safe and effective delivery system can be employed, siRNA has significant potential to improve the efficacy and reduce medication dosage in SARS-CoV-2 infection. Precision therapy is likely to be further improved by targeting specific ligands on infected cells.

We had tried to use specific lipid or peptide modification. Some candidates significantly increased the delivery efficiency to animal lungs, but unfortunately, also increased the immune toxicity. The study is still on-going and under investigation.

Reference

DeVincenzo J, Lambkin-Williams R, Wilkinson T, Cehelsky J, Nochur S, Walsh E, Meyers R, Gollob J, Vaishnav A (2010) A randomized, double-blind, placebo-controlled study of an RNAi-based therapy directed against respiratory syncytial virus. *Proc Natl Acad Sci U S A* 107: 8800-5

Gottlieb J, Zamora MR, Hodges T, Musk AW, Sommerwerk U, Dilling D, Arcasoy S, DeVincenzo J, Karsten V, Shah S, Bettencourt BR, Cehelsky J, Nochur S, Gollob J, Vaishnav A, Simon AR, Glanville AR (2016) ALN-RSV01 for prevention of bronchiolitis obliterans syndrome after respiratory syncytial virus infection in lung transplant recipients. *J Heart Lung Transplant* 35: 213-21

Phillips JE (2017) Inhaled efficacious dose translation from rodent to human: A retrospective analysis of clinical standards for respiratory diseases. *Pharmacol Ther* 178: 141-147

Comment 3:

Page 7, line 5 onwards and Figure 3: The materials and methods part mentions 5 animals per group, but Figures 4E and F only reflect an n of 3. Based on the siRNA amount recovered from the lungs, the average seems comparable for intranasal vs. inhalation delivery, which is not reflected by Figures 3A and 3B. Are the authors certain that the quantification method via stem-loop PCR results in reliable data? Did they perform internal standard experiments with tissues spiked with known amounts of siRNA?

Response

We thank the reviewer for pointing out that the content might confuse the reader. Fig 3A-B and Fig 3E-F represent two independent animal studies for different purposes.

Fig 3A-B

This experiment was conducted in a P3 lab using hACE2 mice (n = 5 per group) treated with C6G25S via AI or IN and followed by virus infection for *in-vivo* efficacy evaluation.

The tissue sections were prepared from half of the tissue at 2 dpi and stained by ISH to visualize the difference in siRNA distribution via inhalation or intranasal instillation. The other half of the tissue was used to quantify the viral titer. C6G25S positive cells were counted and presented in Fig 3D.

Figure 3 (simplified)

Fig 3E–F

This experiment was conducted to understand the different siRNA amounts in the nasal cavities and lungs while delivering siRNA via AI or IN. We carried out an independent experiment using C57/B6 mice (n = 3 per group) operated in a standard animal lab. After mice were treated with siRNA by AI or IN, whole nasal cavities and lungs of tested mice were collected and homogenized to quantify siRNA amount by stem-loop qPCR using a standard curve. The standard curve was conducted using synthesized siRNA spiked into the tissue homogenates of vehicle control mice and showed R² value > 0.99.

Fig 3E shows that the C6G25S concentrations in the lung were 5.8 times higher than that in the nasal cavities when C6G25S was delivered by AI.

Fig 3F showed significant variation in the siRNA amount in lungs among mice while delivering siRNA via IN.

Figure 3 (simplified)

We thank the reviewer for pointing out the misunderstanding in the legend for Fig 3E and 3F. To better distinguish between the two experiments, we have added more details to the description in the legend of Fig 3E and 3F as shown below.

Original statement in Figures legends section (Fig.3 P23 line 8):

E, F. The siRNA levels deposited in lungs and nasal cavities of C57/B6 mice after siRNA delivery via AI (E) and IN (F) (n = 3 in each group), respectively, was quantified. Quantification data represent mean ± SD. P value by Student *t* test.

Revised statement in Figures legends section (Fig.3 P29 line 8):

E, F. C57/B6 mice (n = 3 per group) were treated by AI with C6G25S 1.48 mg/L for 30 min (E) or 50 μL of saline containing 50 μg of C6G25S by IN administration (F). The C6G25S deposited in whole lungs and nasal cavities was quantified after whole tissue homogenization followed by stem-loop RT-qPCR. Quantification data represent mean ± SD. P value by Student *t* test.

Comment 4:

Lung injury was qualitatively investigated by microscopic assessment of lung tissue sections. Rather than a qualitative assessment, a quantitative analysis of cells and cytokines in the lung lavage fluid is suggested.

Response

We appreciate the reviewer's advice.

Because the experiments involved SARS-CoV-2 virus, the infection was conducted in a P3 Lab where the operator is heavily protected. For safety and quality concerns, they do not offer a BAL collection service.

We have added several alternative experiments to answer the reviewer's question, as shown below.

For immune cell analysis in lungs

We performed immunohistochemical staining (IHC) to visualize the distribution of immune cells (including neutrophils, macrophages and lymphocytes) in lungs (Fig 5C–E). Cells were quantified by analyzing the whole tissue section using HALO software (Fig 5G). The results showed that the positively stained area of neutrophil, macrophage and CD3⁺ lymphocytes was reduced by 78.2%, 46.9% and 62.4% by C6G25S treatment, respectively.

Figure 5 (simplified)

For pro-inflammatory cytokine analysis in lungs

We have conducted extra-studies for analyzing gene expression of pro-inflammatory cytokines such as IL-6, IFN- α , TNF- α and IFN- γ in the stored samples. In addition to IFN- α and TNF- α that were undetected, the expression of IL-6 and IFN- γ was

significantly reduced by C6G25S treatment (Fig EV2A). IHC staining for IL-6 (Fig EV2B) and IFN- γ (Fig EV2C) served as verification.

Figure EV2. C6G25S reduced the expression of inflammatory cytokines induced by SARS-CoV-2.

- A.** K18-hACE2-transgenic mice were challenged intranasally with 10^4 PFU of virus and co-treatment with 1.48 mg/L of C6G25S or vehicle control (saline) by AI for 30 min on day 0 (right after infection) and day 1. Mice were sacrificed at day 2 post-infection. Lung tissues of each mouse were collected, and RNA was extracted. The expression of IL-6 and IFN- γ were quantified by RT-PCR. The amount of each cytokine relative to an uninfected sample is presented as mean \pm SD. *P* value by Student *t* test.
- B.** Images of IHC staining of anti-IL6 (Brown color) in lungs of vehicle control (left) and C6G25S treated group (right).
- C.** Images of IHC staining of anti-IFN- γ (Brown color) in lungs of vehicle control (left) and C6G25S treated group (right).

New information has been added to three different sections in the revised manuscript:

(1) Figure EV2 (2) Result section (3) Materials and Methods section

(1) Figure EV2 : Same as above

(2) Result section (page 11 line17)

The pro-inflammatory cytokines including IL-6, IFN- α , TNF- α and IFN- γ were also analyzed. In addition to IFN- α and TNF- α that were under detection limit, the expression of IL-6 and IFN- γ were significantly reduced by C6G25S treatment (**Fig EV2A**). IHC staining for IL-6 (**Fig EV2B**) and IFN- γ (**Fig EV2C**) also confirmed the reduction of cytokines by C6G25S treatment.

(4) Materials and Methods section (page 19 line 43)

Quantification of cytokine mRNA via RT-qPCR

The lungs were homogenized in RLT buffer (Qiagen) using a TissueLyser II (Qiagen) at 4°C and clarified by centrifugation at 15,000 g for 15 min at 4 °C. Total RNA was extracted with a RNeasy Micro Kit (Qiagen) following the manufacturer's instructions. Reverse transcription reaction was conducted with a Maxima First-Strand cDNA Synthesis kit (Thermo Fisher Scientific) and 1 ug of total cellular RNA. qPCR was carried out on a LightCycler 480 using SYBR Green I Master (Roche Diagnostics) with 1:5 dilutions of cDNA. Primer sets used to detect cytokine genes are shown in Appendix Table S7. Each sample was assayed in triplicate to determine an average threshold cycle (Ct) value. Gene expression fold change was calculated using the $\Delta\Delta C_t$ method. The mRNA level of each gene was normalized to constitutively expressed GAPDH mRNA.

Appendix Table S7.

Appendix Table S7. RT-qPCR primers for the quantification of cytokine mRNA

Genes	Forward primer (5'-3')	Reverse primer (5'-3')
IL6	TACCACTTCACAAGTCGGAGGC	CTGCAAGTGCATCATCGTTGTTC
TNF α	GGTGCCTATGTCTCAGCCTCTT	GCCATAGAAGTATGATGAGAGGGAG
IFN γ	CGGCACAGTCATTGAAAGCCTA	GTTGCTGATGGCCTGATTGTC
IFN α	GGATGTGACCTTCTCAGACTC	ACCTTCTCCTGCGGGAATCAA
GAPDH	CGACTTCAACAGCAACTCCACTCTTCC	TGGGTGGTCCAGGGTTTCTTACTCCTT

Antibodies in cytokine staining (page 19 line 33)

IL-6 and IFN- γ were detected using Anti-IL-6 (BS-4539R) and anti-IFN- γ (BS-0480R) purchased from Bioss Company.

Comment 5:

It is not clear why different species (mice, rats and human PMBCs) were used for the same purpose of investigating toxicity and why lung lavage, a very standard technique for quantitative results on toxicity, was not performed in any of the animals.

Response

The single-dose and 14 days repeating-dose toxicity study in rodent is a standard examination for evaluating systemic toxicity of small chemical drugs. Although the lung histopathology had been analyzed, detailed immune responses need to be further characterized.

The human PBMCs experiment is a preliminary assay designed to exclude the siRNA candidates with a risk of stimulating the human immune system.

We appreciate the reviewer's valuable suggestion. Two additional experiments have been conducted to further characterize the immunogenicity of C6G25S.

Study 1

This study showed that the efficacy dose of C6G25S does not stimulate IL-6, TNF- α , IFN- α and IFN- γ expression in ICR mice. In contrast, poly(I:C) and unmodified siRNA are capable of triggering a cytokine response. The data are presented in **Fig EV3**.

Study 2

This study showed that no significant immune cells infiltration in bronchoalveolar lavage or cytokine induction in lung tissue was observed in ICR mice challenged with up to 75 mg/kg of C6G25S. The data are presented in **Fig EV4**.

New information has been added to four different sections in the revised manuscript:

(1) Figure EV3A (2) Figure EV4 (3) Result section (4) Materials and Methods section

(1) Figure EV3A

Figure EV3. C6G25S does not activate non-specific immune response against

SARS-CoV-2.

ICR mice (n = 3) were treated with vehicle alone (saline), 2.5 mg/kg of *poly(I:C)* and negative control siRNA (an unmodified siRNA sequence used in the cell-based virus inhibition assay) via high volume (50 μ L) of intranasal administration. Lung tissue was collected at 48 hours after treatment. The mRNA expression of IL-6, TNF- α , IFN- α and IFN- γ was quantified by RT-PCR. Data are presented as mean \pm SD. *P* value by Student *t* test.

(2) Figure EV4

Figure EV4. No significant immune stimulation was observed in mice treated with high-dose of C6G25S.

- C. ICR mice (n = 3) treated with saline control or different dosages (20, 40, 75 mg/kg) of C6G25S by intranasal instillation. The bronchoalveolar lavage (BAL) was collected at day 2 after treatment. Different WBC cell counts were analyzed by automated hematology analyzer. The cell counts are expressed as mean \pm SD. All of the different treatment groups are statistically insignificant compared with the control.
- D. Lung tissues of each mouse were collected, and RNA was extracted. The mRNA expression of IL-6, TNF- α , IFN- α and IFN- γ was quantified by RT-PCR. The

relative amount of each cytokine is presented as mean \pm SD. All of the different treatment groups are statistically insignificant compared with the control.

(3) Result section (page 12 line 3)

To investigate the potential local immune response of C6G25S, lungs from ICR mice treated with efficacy dose of C6G25S were analyzed for the expression of pro-inflammatory cytokines including IL-6, TNF- α , IFN- α and IFN- γ . C6G25S treatment did not induce any cytokine expression. In contrast, *Poly(I:C)* induced the expression of IL-6, TNF- α and IFN- γ and an extensively used control siRNA (Kermorgant et al., 2004) was shown to induce 8-fold of IL-6 expression (**Fig EV3A**). Despite the induction of IL-6, control siRNA did not show any inhibition of the viral RNA amplification that might be evoked through nonspecific immune responses (**Figure EV3B**). Furthermore, administration up to 75 mg/kg of C6G25S did not induce immune cell infiltration in bronchoalveolar lavage fluid (**Fig EV4A**) and pro-inflammatory cytokines expression in lung tissue (**Fig EV4B**).

(4) Materials and Methods section

Quantification of cytokine mRNA via RT-qPCR (page 19 line 43)

The lungs were homogenized in RLT buffer (Qiagen) using a TissueLyser II (Qiagen) at 4°C and clarified by centrifugation at 15,000 g for 15 min at 4 °C. Total RNA was extracted with a RNeasy Micro Kit (Qiagen) following the manufacturer's instructions. Reverse transcription reaction was conducted with a Maxima First-Strand cDNA Synthesis kit (Thermo Fisher Scientific) and 1 μ g of total cellular RNA. qPCR was carried out on a LightCycler 480 using SYBR Green I Master (Roche Diagnostics) with 1:5 dilutions of cDNA. Primer sets used to detect cytokine genes are shown in Appendix Table S7. Each sample was assayed in triplicate to determine an average threshold cycle (Ct) value. Gene expression fold change was calculated using the $\Delta\Delta$ Ct method. The mRNA level of each gene was normalized to constitutively expressed GAPDH mRNA.

In vitro and in vivo evaluation of immunogenicity of C6G25S (page 20 line 18)

To investigate if C6G25S may cause acute local immune response in the lungs, 7-week-old male Blt \times :CD1(ICR) mice obtained from Shanghai Model Organisms Center (Shanghai, China) were intranasal instilled with vehicle alone (saline), poly IC at 2.5 mg/kg (positive control) or C6G25S at 0, 20, 40, or 75 mg/kg. After 48 h, mice were euthanized by isoflurane inhalation and bronchoalveolar lavage fluid (BALF) collected via intratracheal infusion of 0.8 mL of PBS. BALF was obtained by retracting the piston of the syringe three times and centrifuged at 450 g for 5 min at 4 °C. The resulting cell pellet was resuspended in 200 μ L of PBS and analyzed immediately using an automated hemacytometer (BX3010, Sysmex). The supernatant was stored at -80 °C. The mRNA levels of cytokines in lungs were determined via RT-qPCR. NC siRNA (Kermorgant et al., 2004) and NC siRNA2 were used as controls. NC siRNA2 was designed by blasting the database for no match to human sequences and SARS-CoV-2 genome. The sense and antisense sequence of NC siRNA was 5'-UUC UCC GAA CGU GUC ACG UTT-3' and 5'-ACG UGA CAC GUU CGG AGA ATT-3'. NC siRNA2 was 5'-UUC GAC CGG UAU AUG GUA GTT-3' and 5'-CUA CCA UAU ACC GGU CGA ATT-3'.

Appendix Table S7.

Appendix Table S7. RT-qPCR primers for the quantification of cytokine mRNA

Genes	Forward primer (5'-3')	Reverse primer (5'-3')
IL6	TACCACTTCACAAGTCGGAGGC	CTGCAAGTGCATCATCGTTGTTC
TNF α	GGTGCCTATGTCTCAGCCTCTT	GCCATAGAAGTATGATGAGAGGGAG
IFN γ	CGGCACAGTCATTGAAAGCCTA	GTTGCTGATGGCCTGATTGTC
IFN α	GGATGTGACCTTCCTCAGACTC	ACCTTCTCCTGCGGGAATCCAA
GAPDH	CGACTTCAACAGCAACTCCCCTCTTCC	TGGGTGGTCCAGGGTTTCTACTCCTT

Reference

Kermorgant S, Zicha D, Parker PJ (2004) PKC controls HGF-dependent c-Met traffic, signalling and cell migration. *EMBO J* 23: 3721-34

Comment 6:

On page 4, line 11ff the authors write:

Further evaluating the location of the siRNA binding sites on the vital genes involved in virus replication and infection, 374 located in regions encoding the viral leader, papain-like protease, 3C-like protease, RNA dependent RNA polymerase (RdRp), helicase, spike protein, and the envelope protein were isolated. Due to the specific (discontinuous) replication of coronaviruses, all viral transcripts carry a common 5' (Leader Sequence) and 3' end (N/ORF10 and 3'UTR) which was also confirmed for SARS-CoV-2 (e.g. <https://doi.org/10.1016/j.cell.2020.04.011>). As the siRNA targets the viral RNAs and not proteins, it is not possible to specifically target the indicated genes. Thus, the described procedure to me makes no sense.

Response

We thank the reviewer for pointing out that the content might confuse the readers. It is indeed that discontinuous transcription occurs to produce a set of nested 3' and 5' co-terminal subgenomic RNAs (sgRNAs), when the copy of the TRS-B hybridizes with the TRS-L. We have revised the related content in the Result section and compared as below:

Original statement in Result section (page 5):

Next, 674 siRNA candidates with over 99.8% coverage rate among 29,871 SARS-CoV-2 genomes and their corresponding targeting regions with low propensity for secondary structure were selected (Lan, Allan et al., 2020, Rangan, Zheludev et al., 2020). Further evaluating the location of the siRNA binding sites on the vital genes involved in virus replication and infection, 374 located in regions encoding the viral leader, papain-like protease, 3C-like protease, RNA dependent RNA polymerase (*RdRp*), helicase, spike protein, and the envelope protein were isolated.

Revised the statement in Result section (page 5 line 14):

Next, 674 siRNA candidates with over 99.8% coverage rate among 29,871 SARS-CoV-2 genomes and their corresponding targeting regions with low propensity for

secondary structure were selected (Lan et al., 2020, Rangan et al., 2020). Furthermore, 374 siRNA candidates targeting to different regions of SARS-CoV-2 genomes were selected, including those in leader, papain-like protease, 3C-like protease, RNA dependent RNA polymerase (*RdRp*), helicase, spike, and the envelope coding regions (Kim, Lee et al., 2020).

Reference

Lan TCT, Allan MF, Malsick LE, Khandwala S, Nyeo SSY, Bathe M, Griffiths A, Rouskin S (2020) Structure of the full SARS-CoV-2 RNA genome in infected cells. 2020.06.29.178343

Rangan R, Zheludev IN, Hagey RJ, Pham EA, Wayment-Steele HK, Glenn JS, Das R (2020) RNA genome conservation and secondary structure in SARS-CoV-2 and SARS-related viruses: a first look. *RNA* 26: 937-959

Kim D, Lee JY, Yang JS, Kim JW, Kim VN, Chang H (2020) The Architecture of SARS-CoV-2 Transcriptome. *Cell* 181: 914–921

Comment 7:

The authors in many instances appear to overstate the role of escape mutations:

a) E.g. in the last paragraph in the discussion they write "vaccine-resistant variants continue to..." After all the most widely used vaccines strongly reduce the mortality and hospitalization also by the Delta variant. Also, even in case that new and more relevant escape variants should occur, most of the vaccines could also be quickly adapted (e.g. mRNA vaccines). The authors should thus downtone the statements on the limitations of the vaccines and the role of the delta variant.

b) On page 3 line 6ff the authors write: According to the United States Centers for Disease Control and Prevention, 90% of 469 new infections in Barnstable County, Massachusetts were caused by the Delta variant, and among those, 74% were already fully vaccinated (Brown, Vostok et al., 2021).

I strongly believe that such a presentation is misleading. The data could only be fully interpreted if one would know the fraction of vaccinated people.

Response

We appreciate the reviewer's advice. We have modified the content in the Introduction section as shown below.

Original statement in Introduction section (page 3):

According to the United States Centers for Disease Control and Prevention, 90% of 469 new infections in Barnstable County, Massachusetts were caused by the Delta variant, and among those, 74% were already fully vaccinated (Brown, Vostok et al., 2021). Furthermore, vaccinated and unvaccinated people carry similar viral loads. These data suggest that the current vaccine strategy fails to halt SARS-CoV-2 transmission. The current protein-based therapeutics (i.e. vaccines, antibodies, or

convalescent plasma) primarily target the spike protein, which is the main protruding structure on the virus's surface. However, the efficacy of such approaches might not last over a long term due to the highly-mutated feature of spike protein (van Dorp, Acman et al., 2020).

Revised the statement in Introduction (page 3 line 6):

The SARS-CoV-2 vaccines initially achieved great success in reducing viral infection and severe illness. The effectiveness of ChAdOx1 nCoV-19, BNT162b2 and mRNA-1273 vaccine reached 70.4%, 95% and 94.1%, respectively (Baden, El Sahly et al., 2021, Knoll & Wonodi, 2021, Wang, 2021). Nevertheless, the emergence of new variants, especially the Beta and Delta variants, has raised great concerns, since reduced sensitivity of SARS-CoV-2 variants to therapeutic neutralizing antibodies, serum from convalescent patients and vaccinated individuals have been reported (Planas, Veyer et al., 2021). In addition, vaccine breakthrough cases have been described. A report from the UK indicates that the effectiveness of two doses of ChAdOx1 nCoV-19 vaccine against delta infection reduces to 67% while BNT162b2 reduces to 88% (Lopez Bernal, Andrews et al., 2021). Another report from Qatar showed that two doses of BNT162b2 only present 51.9% of effectiveness while mRNA-1273 presents 73.1% (Tang, Hasan et al., 2021). These reports implicate that viral mutations might influence vaccine effectiveness. Recently, a novel variant called Omicron, that contains twice the number of mutations in the spike protein compared with the Delta variant, has raised further concerns (Gao, Guo et al., 2021). Although the real-world effectiveness of vaccines against Omicron infection has not yet been released, reduced neutralizing activity of the Omicron variant against mRNA vaccine-induced antibody responses has been reported (Edara, Manning et al., 2021), and a 3rd dose of vaccination is currently suggested to provide robust neutralizing antibody responses against the Omicron variant. The emergence of Omicron variant indicates how fast the SARS-CoV-2 evolves, and its potential impact on the current protein-based intervention (i.e., vaccines, antibodies, or convalescent plasma) that primarily target the highly-mutated spike protein (van Dorp et al., 2020) can not be neglected.

Reference

Baden LR, El Sahly HM, Essink B, Kotloff K, Frey S, Novak R, Diemert D, Spector SA, Roupheal N, Creech CB, McGettigan J, Khetan S, Segall N, Solis J, Brosz A, Fierro C, Schwartz H, Neuzil K, Corey L, Gilbert P et al. (2021) Efficacy and Safety of the mRNA-1273 SARS-CoV-2 Vaccine. *N Engl J Med* 384: 403-416

Edara V-V, Manning KE, Ellis M, Lai L, Moore KM, Foster SL, Floyd K, Davis-Gardner ME, Mantus G, Nyhoff LE, Bechnak S, Alaaeddine G, Naji A, Samaha H, Lee M, Bristow L, Hussaini L, Ciric CR, Nguyen P-V, Gagne M et al. (2021) mRNA-1273 and BNT162b2 mRNA vaccines have reduced neutralizing activity against the SARS-CoV-2 Omicron variant. 2021.12.20.473557

Gao SJ, Guo H, Luo G (2021) Omicron variant (B.1.1.529) of SARS-CoV-2, a global urgent public health alert! *J Med Virol* 2021 Nov 30. doi: 10.1002/jmv.27491.

Knoll MD, Wonodi C (2021) Oxford-AstraZeneca COVID-19 vaccine efficacy. *Lancet* 397: 72-74

Lopez Bernal J, Andrews N, Gower C, Gallagher E, Simmons R, Thelwall S, Stowe J, Tessier E, Groves N, Dabrera G, Myers R, Campbell CNJ, Amirthalingam G,

Edmunds M, Zambon M, Brown KE, Hopkins S, Chand M, Ramsay M (2021) Effectiveness of Covid-19 Vaccines against the B.1.617.2 (Delta) Variant. *N Engl J Med* 385: 585-594

Planas D, Veyer D, Baidaliuk A, Staropoli I, Guivel-Benhassine F, Rajah MM, Planchais C, Porrot F, Robillard N, Puech J, Prot M, Gallais F, Gantner P, Velay A, Le Guen J, Kassis-Chikhani N, Edriss D, Belec L, Seve A, Courtellemont L et al. (2021) Reduced sensitivity of SARS-CoV-2 variant Delta to antibody neutralization. *Nature* 596: 276-280

Tang P, Hasan MR, Chemaitelly H, Yassine HM, Benslimane FM, Al Khatib HA, AlMukdad S, Coyle P, Ayoub HH, Al Kanaani Z, Al Kuwari E, Jeremijenko A, Kaleeckal AH, Latif AN, Shaik RM, Abdul Rahim HF, Nasrallah GK, Al Kuwari MG, Al Romaihi HE, Butt AA et al. (2021) BNT162b2 and mRNA-1273 COVID-19 vaccine effectiveness against the SARS-CoV-2 Delta variant in Qatar. *Nat Med* 27, pages2136–2143

van Dorp L, Acman M, Richard D, Shaw LP, Ford CE, Ormond L, Owen CJ, Pang J, Tan CCS, Boshier FAT, Ortiz AT, Balloux F (2020) Emergence of genomic diversity and recurrent mutations in SARS-CoV-2. *Infect Genet Evol* 83: 104351

Wang X (2021) Safety and Efficacy of the BNT162b2 mRNA Covid-19 Vaccine. *N Engl J Med* 384: 1577-1578

Comment 8:

On page 3 line 8ff they write: Furthermore, vaccinated and unvaccinated people carry similar viral loads. These data suggest that the current vaccine strategy fails to halt SARS-CoV-2 transmission. What do the authors want to achieve with their siRNA therapy? Which is the patient group they are planing to use the drug for? Halt transmission in otherwise asymptomatic people or people with mild symptoms?

Response

We appreciate the reviewer’s valuable question. We had rewritten the introduction section and deleted this description.

For our clinical design, we plan to initially apply our drug to treat patients with mild symptoms first. Early administrations of siRNA against SARS-CoV-2 after the onset of symptoms might reduce viral load in the early stage and alleviate the severity of subsequent infection and inflammation in the lungs.

Subsequently, we will not exclude the possibility of expanding the trial to treat asymptomatic infected people for accelerating the clearance of the virus or pre-treat healthy individuals for reducing the risk of infection.

However, the plan could always be changed depending on the situation of the pandemic or after clinical data has been obtained.

Comment 9:

What is the goal of the transcriptome analysis? While I understand that it is of interest to identify potential off-target genes, I believe it is very hard to say that a low number of hits in their analysis corresponds to low off-target activity. E.g. it appears to me that the authors have set the threshold to define a significant de-regulated gene very high (threefold difference, FDR 0.001). Also, I cannot find any information on the number of biological replicates they analyzed, which would also influence the number of genes which one would find de-regulated

Thus, while I see the purpose of this analysis to define potential off-targets, I do not see that a low number of significant hits (by their definition) few off-target activity

Response

We appreciate the reviewer's valuable advice.

The US FDA has suggested us to collect off-target data for evaluating the possible risk during a pre-IND meeting. We agreed that using three-fold difference was too high and the related description was not clear. Thus, we have conducted a biological replicate experiment, re-analyzed our data and set two-fold difference as the cut-off.

The **Result section** and **Appendix Fig S1** were revised as shown below.

New information was added to two different sections in the revised manuscript: (1)Result section (2) Appendix Fig. S1

(1) Result section

Original statement in Result section (page 5 line 8):

Whole transcriptome analysis using next generation sequencing (NGS) showed that the modification of C6 reduced the total number of off-target genes in BEAS-2B cells from 21 to 15 (**Appendix Fig S1**). The 15 off-target genes were further verified by RT-qPCR and only four genes, including *CXCL5*, *REEP3*, *SGPPI1*, and *ARTN*, were confirmed to be true off-target genes (**Table 2**).

Revised statement in Result section (page 6 line 13):

To evaluate the potential siRNA off-target effect, we performed two independent biological replicates of whole transcriptome sequencing from BEAS-2B cells transfected with the unmodified C6 and fully modified C6G25S, respectively. Whole transcriptome analysis showed that the modification of C6 could significantly reduce the total number of off-target genes in BEAS-2B cells from 51 (C6) to 21 (C6G25S) for two-fold expression differences (**Appendix Fig S1**). The 21 genes identified with two-fold expression differences after C6G25S treatment are listed in Table 2. Top 10 genes were confirmed by RT-qPCR.

(2) Appendix Fig. S1

Revised Appendix Fig. S1:

Appendix Fig. S1:

Modification of C6 significantly reduced off-targets analyzed by RNA-seq.

Scatterplots display global gene expression change in C6- and C6G25S-treated BEAS-2B cells compared with no siRNA control, respectively. Red dots indicate genes down-regulated with fold change ≥ 2 . *TPM: Transcripts per Million.

Comment 10:

There is in general very few information on the use of controls.

- In their cell culture experiment shown in Figure 1D,E,F and Figure 2B: Were the values set in relation to a control siRNA?
- Also, I cannot find any information in the figure legend of Figures describing the *in vivo* data: E.g. in Figure 4 and Figure 5F,H what the control consisted of. Is this a non-relevant or scrambled siRNA? Or an untreated animal? This would be an important information to judge if the therapy was specific or if it could have been a result of an unspecific immunostimulatory effect.

Response

We thank the reviewer for pointing out the omission. We used a non-relevant siRNA (Negative control, NC siRNA) in cell-based experiments, and the sequence description of NC siRNA has been provided in the method section (siRNA screening in Vero E6 cells).

In all animal experiments, we used vehicle alone as the control group. We have added the control information for every in-vivo experiment in the legends of Figs

3, 4 and 5 as well as in the Method section of the revised manuscript.

The reasons we used vehicle alone as control in the animal study are:

1. C6G25S is a fully modified siRNA and relatively non-immunogenic based on our study.
2. NC siRNA is unmodified siRNA that could be immunogenic and induce nonspecific immunostimulatory effects as shown in Fig EV3.

Moreover, we have conducted a small study in a P3 lab to show that NC siRNA is incapable of significantly inhibiting SARS-CoV-2 replication (Figure EV3B).

New information was added to two different sections in the revised manuscript: (1) Figure EV3 (2) Materials and Methods section

(1) Figure EV3

Figure EV3. C6G25S does not activate non-specific immune response against SARS-CoV-2.

- A. ICR mice (n = 3) were treated with vehicle alone (saline), 2.5 mg/kg of poly(I:C) and negative control siRNA (an unmodified siRNA sequence used in the cell-based virus inhibition assay) via high volume (50 μ L) of intranasal administration. Lung tissue was collected at 48 hours after treatment. The mRNA expression of IL-6, TNF- α , IFN- α and IFN- γ was quantified by RT-PCR.
- B. hACE2 transgenic mice (n = 3) were pre-treated with vehicle alone, 2.5 mg/kg of NC siRNA, NC siRNA 2 and C6G25S via high volume (50 μ L) of intranasal

administration at 48 hours before virus infection. Viral RNA (left) and infectious virions (right) in lungs were quantified with RT-qPCR and plaque forming assay, respectively, at 2 days post-infection.

Data are presented as mean \pm SD. *P* value by Student *t* test.

(2) Materials and Methods section (page 20 line 19)

To investigate if C6G25S may cause acute local immune response in the lungs, 7-week-old male Blt^w:CD1(ICR) mice obtained from Shanghai Model Organisms Center (Shanghai, China) were intranasal instilled with vehicle alone (saline), poly IC at 2.5 mg/kg (positive control) or C6G25S at 0, 20, 40, or 75 mg/kg. After 48 h, mice were euthanized by isoflurane inhalation and bronchoalveolar lavage fluid (BALF) collected via intratracheal infusion of 0.8 mL of PBS. BALF was obtained by retracting the piston of the syringe three times and centrifuged at 450 g for 5 min at 4 °C. The resulting cell pellet was resuspended in 200 μ L of PBS and analyzed immediately using an automated hemacytometer. The supernatant was stored at -80 °C. The mRNA levels of cytokines in lungs were determined via RT-qPCR. NC siRNA (Kermorgant et al., 2004) and NC siRNA2 were used as controls. NC siRNA2 was designed by blasting the database for no match to human sequences and SARS-CoV-2 genome. The sense and antisense sequence of NC siRNA was 5'-UUC UCC GAA CGU GUC ACG UTT-3' and 5'-ACG UGA CAC GUU CGG AGA ATT-3'. NC siRNA2 was 5'-UUC GAC CGG UAU AUG GUA GTT-3' and 5'-CUA CCA UAU ACC GGU CGA ATT-3'.

Reference

Kermorgant S, Zicha D, Parker PJ (2004) PKC controls HGF-dependent c-Met traffic, signalling and cell migration. *EMBO J* 23: 3721-34

Comment 11:

Page 3 lines 3/4

(University, 2021) , please check if the citation is displayed in a correct way

Response

We thank the reviewer for pointing out this inaccuracy of citation. The description has been modified as shown below:

Original statement in P3 line 3:

Severe acute respiratory syndrome coronavirus 2 (SARS-CoV-2) has infected over 200 million people and caused more than 4.5 million deaths worldwide as of September 13, 2021(University, 2021).

Revised statement in P3 line 2:

Severe acute respiratory syndrome coronavirus 2 (SARS-CoV-2) has infected over 200 million people and caused more than 4.5 million deaths worldwide as of September 13, 2021, according to the John Hopkins coronavirus resource center.

Response for Reviewer 3:

The development of new therapeutics against SARS-CoV-2 variants is an important topic. Authors developed an inhalable siRNA that inhibit SARS-CoV-2 VOCs at picomolar ranges *in vitro* and is active in ACE2-mice. Experiments were done with infectious virus under BSL3 conditions, the siRNA is probably the first to shown to be active *in vivo* against CoV2, without causing severe side effects.

We greatly appreciated the **Reviewer 3** for acknowledging the value of our study. We have further improved the manuscript by incorporating **Reviewer 3**'s suggestion. The point-by-point responses are provided directly afterward.

Comment 1:

Authors claim to have performed a "post-exposure treatment" but actually it looks as if the drug and the virus were administered simultaneously. This must be clarified throughout the manuscript. Post exposure implies that virus has already established infection, and this seems not to be the case.

Response

We thank the reviewer for pointing out that it might be inappropriate to use postexposure treatment in our case.

Details of our postexposure treatment as presented below:

Mice were first anesthetized with Zoletil/Dexdomitor and infected intranasally with SARS-CoV-2. After 30 mins of recovery, mice were placed in the chamber for 30 min of inhalation treatment.

Given that the interval between siRNA delivery and virus infection was short, we have replaced the "postexposure treatment" with "co-treatment" as advised.

We have modified all the descriptions of postexposure treatment to co-treatment. In total, there are 14 sites identified in the Main text and 0 in the Appendix as listed below:

Site	Original	Revised
P1 Line 8	post-exposure treatment in K18-hACE2-transgenic mice	co-treatment in K18-hACE2-transgenic mice
P7 Line 21	post-exposure treatment in vivo.	co-treatment in vivo.
P8 Line 2	postexposure administration of C6G25S	co-treatment of C6G25S
P8 Line 5	96.2% in the postexposure group	96.2% in the co-treatment group
P8 Line 7	96% was observed in the postexposure group	96% was observed in the co-treatment group
P8 Line 13	Two post-exposure groups	Two co-treatment groups
P8 Line 19	prophylactic and post-exposure treatment	prophylactic and co-treatment
P11 Line 16	In the post-exposure treatment	In the co-treatment
P13 Line 46	for postexposure treatment.	for co-treatment.
P14 Line 34	For postexposure treatment	For co-treatment
P18 Line 19	post-exposure treatment in hACE2 transgenic mice.	co-treatment in hACE2 transgenic mice.
P23 Line 11	Fig. 4. Prophylatic and post-exposure	Fig. 4. Prophylatic and co-treatment
P23 Line 18	postexposure treated with	co-treatment with
P23 Line 23	Postexposure treatment of C6G25S	Co-treatment of C6G25S

Comment 2:

What are the disadvantages or limitations of RNAs as inhalative drugs?

Response

We thank the reviewer for their valuable consideration. We have raised the related discussion about the disadvantage or limitations of siRNA as inhalable drugs in page 15 (Discussion section).

New statement in Discussion section (page 15 line 12)

In conclusion, the promising efficacy of respiratory-delivered naked C6G25S has been demonstrated in this report. However, in spite of picomolar range of IC_{50} *in vitro*, a relatively high administered dose of C6G25S is still needed. If a safe and effective delivery system can be employed, C6G25S has significant potential to improve the efficacy and reduce medication dosage. Precision therapy is likely to be further improved by targeting specific ligands on infected cells.

Are there similar approaches against other (respiratory) viruses?

Response

We thank for the reviewer's valuable consideration. We have raised the related discussion about the approaches against other respiratory viruses in page 4 (Introduction section) and page 13 (Discussion section).

New statement in Introduction section (page 4 line 9)

Since Bitko et al. reported that intranasal instillation (IN) of unmodified naked siRNA is capable of inhibiting respiratory viral infection in mice without the help of any carrier or transfection reagent (Bitko, Musiyenko et al., 2005), IN and aerosol inhalation (AI) of naked siRNA have been widely utilized for delivering siRNA into pulmonary cells (Kandil & Merkel, 2019, Zafra, Mazzeo et al., 2014). One successful application of an unmodified naked siRNA is ALN-RSV01 administered by nasal spray or aerosol inhalation to treat respiratory syncytial virus (RSV) infection. The phase II clinical trial showed that pretreatment with ALN-RSV01 significantly reduces the prevalence of RSV infection (DeVincenzo, Lambkin-Williams et al., 2010) and posttreatment with ALN-RSV01 reduces the risk of bronchiolitis obliterans syndrome in RSV-infected lung transplant patients (Gottlieb, Zamora et al., 2016). These findings suggest the possibility of using respiratory-delivered, unmodified naked siRNA against SARS-CoV-2 infection. By targeting a highly conserved region of SARS-CoV-2, siRNA is capable of inhibiting a wide-spectrum of viral variants and could thus be a one-for-all therapy for the rapidly evolving SARS-CoV-2.

New statement in Discussion section (page 13 line 8)

IN delivery of naked siRNA has been proven to be an effective approach to prevent and treat SARS-CoV-1 infection in non-human primates (Rhesus macaque) (Li et al., 2005). Because the SARS-CoV-1 outbreak had been brought under control in a short period of time, the therapeutics has not proceeded to clinical use. Recently, several studies testing siRNA against SARS-CoV-2 infection have been published (Niktab et al., 2021, Shawan et al., 2021, Tolksdorf et al., 2021, Wu & Luo, 2021). While most

researchers have pursued a siRNA design that has been validated through *in vitro* cell-based experiments, only two publications have assessed the efficacy of their siRNA in an animal model. One publication used intravenous administration of LNPs-siRNA to inhibit SARS-CoV-2 infection in hACE2 transgenic mice (Idris et al., 2021), the viral titer was reduced by about a log of magnitude at day 3 post-infection by prophylactic treatment. The other study used positive-charged dendrimer to carry siRNA and treated SARS-CoV-2-infected Syrian hamster via inhalation (Khaitov et al., 2021). The viral titer was reduced by 30% at day 2 post-infection. The potency of viral inhibition reported in these studies was not as significant as ours. Moreover, LNPs and positive charge dendrimer have been found to be capable of inducing immune or cell toxicity (Kedmi et al., 2010, Kharwade et al., 2021), which might limit the safety window for dosing. These properties also increase the difficulty and cost of industrial production. On the contrary, the feasibility and safety of respiratory-delivered, unmodified naked siRNA has been demonstrated in animal models and in clinical trials (DeVincenzo et al., 2010, Li et al., 2005). Taken together, fully modified C6G25S with low immunogenicity (Figure EV3A and EV4) and reduced off-target effect (**Appendix Fig S1**) was developed for naked delivery to the respiratory system as a safe, effective and feasible approach against SARS-CoV-2.

Reference

Bitko V, Musiyenko A, Shulyayeva O, Barik S (2005) Inhibition of respiratory viruses by nasally administered siRNA. *Nat Med* 11: 50-5

DeVincenzo J, Lambkin-Williams R, Wilkinson T, Cehelsky J, Nochur S, Walsh E, Meyers R, Gollob J, Vaishnav A (2010) A randomized, double-blind, placebo-controlled study of an RNAi-based therapy directed against respiratory syncytial virus. *Proc Natl Acad Sci U S A* 107: 8800-5

Gottlieb J, Zamora MR, Hodges T, Musk AW, Sommerwerk U, Dilling D, Arcasoy S, DeVincenzo J, Karsten V, Shah S, Bettencourt BR, Cehelsky J, Nochur S, Gollob J, Vaishnav A, Simon AR, Glanville AR (2016) ALN-RSV01 for prevention of bronchiolitis obliterans syndrome after respiratory syncytial virus infection in lung transplant recipients. *J Heart Lung Transplant* 35: 213-21

Kandil R, Merkel OM (2019) Pulmonary delivery of siRNA as a novel treatment for lung diseases. *Ther Deliv* 10: 203-206

Kedmi R, Ben-Arie N, Peer D (2010) The systemic toxicity of positively charged lipid nanoparticles and the role of Toll-like receptor 4 in immune activation. *Biomaterials* 31: 6867-75

Kharwade R, Badole P, Mahajan N, More S (2021) Toxicity And Surface Modification Of Dendrimers: A Critical Review. *Curr Drug Deliv*. DOI: 10.2174/1567201818666211021160441

Idris A, Davis A, Supramaniam A, Acharya D, Kelly G, Tayyar Y, West N, Zhang P, McMillan CLD, Soemardy C, Ray R, O'Meally D, Scott TA, McMillan NAJ, Morris KV (2021) A SARS-CoV-2 targeted siRNA-nanoparticle therapy for COVID-19. *Mol Ther* 29: 2219-2226

Khaitov M, Nikonova A, Shilovskiy I, Kozhikhova K, Kofiadi I, Vishnyakova L, Nikolskii A, Gattinger P, Kovchina V, Barvinskaia E, Yumashev K, Smirnov V,

Maerle A, Kozlov I, Shatilov A, Timofeeva A, Andreev S, Koloskova O, Kuznetsova N, Vasina D et al. (2021) Silencing of SARS-CoV-2 with modified siRNA-peptide dendrimer formulation. *Allergy* 76: 2840-2854

Li BJ, Tang Q, Cheng D, Qin C, Xie FY, Wei Q, Xu J, Liu Y, Zheng BJ, Woodle MC, Zhong N, Lu PY (2005) Using siRNA in prophylactic and therapeutic regimens against SARS coronavirus in Rhesus macaque. *Nat Med* 11: 944-51

Meng Z, Lu M (2017) RNA Interference-Induced Innate Immunity, Off-Target Effect, or Immune Adjuvant? *Front Immunol* 8: 331

Niktab I, Haghparast M, Beigi MH, Megraw TL, Kiani A, Ghaedi K (2021) Design of advanced siRNA therapeutics for the treatment of COVID-19. *Meta Gene* 29: 100910

Robbins M, Judge A, Liang L, McClintock K, Yaworski E, MacLachlan I (2007) 2'-O-methyl-modified RNAs act as TLR7 antagonists. *Mol Ther* 15: 1663-9

Shawan M, Sharma AR, Bhattacharya M, Mallik B, Akhter F, Shakil MS, Hossain MM, Banik S, Lee SS, Hasan MA, Chakraborty C (2021) Designing an effective therapeutic siRNA to silence RdRp gene of SARS-CoV-2. *Infect Genet Evol* 93: 104951

Tolksdorf B, Nie C, Niemeyer D, Rohrs V, Berg J, Lauster D, Adler JM, Haag R, Trimpert J, Kaufer B, Drosten C, Kurreck J (2021) Inhibition of SARS-CoV-2 Replication by a Small Interfering RNA Targeting the Leader Sequence. *Viruses* 13: 10. DOI: 10.3390/v13102030

Wu R, Luo KQ (2021) Developing effective siRNAs to reduce the expression of key viral genes of COVID-19. *Int J Biol Sci* 17: 1521-1529

Zafra MP, Mazzeo C, Gamez C, Rodriguez Marco A, de Zulueta A, Sanz V, Bilbao I, Ruiz-Cabello J, Zubeldia JM, del Pozo V (2014) Gene silencing of SOCS3 by siRNA intranasal delivery inhibits asthma phenotype in mice. *PLoS One* 9: e91996

Comment 3:

Title: Remove "C6G25S"

Response

We appreciate the reviewer's advice. We have modified the title as shown below:

Original title in page 1:

A C6G25S siRNA, targets and inhibits a broad range of SARS-CoV-2 infections including Delta variant

Revised title in page 1:

A siRNA targets and inhibits a broad range of SARS-CoV-2 infections including Delta variant

Comment 4:

abstract emergence of SARS-2 (add-2)

Response

We appreciate the reviewer's advice. We have modified the sentence as shown below:

Original statement in P2 line 1:

The emergence of **severe acute respiratory syndrome coronavirus** variants has altered the trajectory of the COVID-19 pandemic and raised some uncertainty on long term efficiency of vaccine strategy.

Revised statement in P2 line 1:

The emergence of **severe acute respiratory syndrome coronavirus-2 (SARS-CoV-2)** variants has altered the trajectory of the COVID-19 pandemic and raised some uncertainty on long term efficiency of vaccine strategy.

Comment 5:

page 3 line 10: "fails to" is too strong statmenet

Response

We appreciate the reviewer's advice. We have rewritten the introduction to tone down the statement and incorporate Omicron variant as shown below:

Revised statement in P3 line 6:

The SARS-CoV-2 vaccines initially achieved great success in reducing viral infection and severe illness. The effectiveness of ChAdOx1 nCoV-19, BNT162b2 and mRNA-1273 vaccine reached 70.4%, 95% and 94.1%, respectively (Baden et al., 2021, Knoll & Wonodi, 2021, Wang, 2021). Nevertheless, the emergence of new variants, especially the Beta and Delta variants, has raised great concerns, since reduced sensitivity of SARS-CoV-2 variants to therapeutic neutralizing antibodies, serum from convalescent patients and vaccinated individuals have been reported (Planas et al., 2021). In addition, vaccine breakthrough cases have been described. A report from the UK indicates that the effectiveness of two doses of ChAdOx1 nCoV-19 vaccine against delta infection reduces to 67% while BNT162b2 reduces to 88% (Lopez Bernal et al., 2021). Another report from Qatar showed that two doses of BNT162b2 only present 51.9% of effectiveness while mRNA-1273 presents 73.1% (Tang et al., 2021). These reports implicate that viral mutations might influence vaccine effectiveness. Recently, a novel variant called Omicron, that contains twice the number of mutations in the spike protein compared with the Delta variant, has raised further concerns (Gao et al., 2021). Although the real-world effectiveness of vaccines against Omicron infection has not yet been released, reduced neutralizing activity of the Omicron variant against mRNA vaccine-induced antibody responses has been reported (Edara et al., 2021), and a 3rd dose of vaccination is currently suggested to provide robust neutralizing antibody responses against the Omicron

variant. The emergence of Omicron variant indicates how fast the SARS-CoV-2 evolves, and its potential impact on the current protein-based intervention (i.e., vaccines, antibodies, or convalescent plasma) that primarily target the highly-mutated spike protein (van Dorp et al., 2020) can not be neglected.

Comment 6:

page 3 line 11: vaccines are not therapeutics

Response

We appreciate the reviewer's advice. We have modified the sentence as shown below:

Original statement in P3 line 11:

protein-based **therapeutics** (i.e. vaccines, antibodies, or convalescent plasma)

Revised statement in P4 line 2:

protein-based **interventions** (i.e., vaccines, antibodies, or convalescent plasma)

Comment 7:

page 4 line 20. Why "E" selected for amplification?

Response

We appreciate the reviewer's valuable consideration.

WHO recommends an initial screening with the E gene (Corman, Landt et al., 2020) and we found that targeting E gene for the determination of viral amplification is more sensitive than any other gene (Nalla, Casto et al., 2020). Therefore, our standard procedure to quantify viral RNA copies is conducted via E gene-based RT-qPCR.

To confirm the inhibition effect of C6G25S on its direct target (RdRp mRNA), we also performed RT-qPCR targeting the RdRp gene as shown in Fig 1F. The IC₅₀ calculated from RdRp-based RT-qPCR (0.13 nM) is quite close to that from E gene-based RT-qPCR (0.17 nM).

Reference

Corman VM, Landt O, Kaiser M, Molenkamp R, Meijer A, Chu DK, Bleicker T, Brunink S, Schneider J, Schmidt ML, Mulders DG, Haagmans BL, van der Veer B, van den Brink S, Wijsman L, Goderski G, Romette JL, Ellis J, Zambon M, Peiris M et al. (2020) Detection of 2019 novel coronavirus (2019-nCoV) by real-time RT-PCR. *Euro Surveill* 25

Nalla AK, Casto AM, Huang MW, Perchetti GA, Sampoleo R, Shrestha L, Wei Y, Zhu H, Jerome KR, Greninger AL (2020) Comparative Performance of SARS-CoV-2

Comment 8:

page 5 line 12: what is "true" meaning?

Response

We appreciate the reviewer pointing out the ambiguous description about the off-target genes.

Because the off-target genes identified by transcriptome analysis need to be verified by RT-qPCR, the true was used to describe the confirmed off-target genes and to distinguish that from the false off-target signal.

We had been advised to perform additional experiment and re-analyzed the off-target data. Detailed description regarding the evaluation of off-target genes caused by the unmodified C6 and fully modified C6G25S are provided in the **Result section** as shown below.

Original statement in P5 line 8:

Whole transcriptome analysis using next generation sequencing (NGS) showed that the modification of C6 reduced the total number of off-target genes in BEAS-2B cells from 21 to 15 (Appendix Fig S1). The 15 off-target genes were further verified by RT-qPCR and only four genes, including CXCL5, REEP3, SGPP1, and ARTN, were confirmed to be true off-target genes (Table 2).

Revised statement in P6 line 13:

To evaluate the potential siRNA off-target effect, we performed two independent biological replicates of whole transcriptome sequencing from BEAS-2B cells transfected with the unmodified C6 and fully modified C6G25S, respectively. Whole transcriptome analysis showed that the modification of C6 could significantly reduce the total number of off-target genes in BEAS-2B cells from 51 (C6) to 21 (C6G25S) for two-fold expression differences (Appendix Fig S1). The 21 genes identified with two-fold expression differences after C6G25S treatment are listed in Table 2. Top 10 genes were confirmed by RT-qPCR.

Comment 9:

page 5 line 13: rephrase, C6 does not inhibit the E gene but amplification of viral RNA using primer binding to E?

Response

We thank the reviewer for pointing out that the description might confuse the reader. We have modified the sentence in the Result section (page 6) and the Y axis title of Figs 1E and 2B as shown below.

New information was added to three different sections in the revised manuscript: (1) Result section (2) Figure 1E (3) Figure 2B

(1) Result section

Original statement in P5 line 13:

Moreover, C6G25S and unmodified C6 were found to have a similar IC₅₀ for inhibiting the viral envelope gene. (0.17 and 0.18 nM, respectively) (Fig 1E).

Revised statement in P6 line 20:

Moreover, C6G25S and unmodified C6 were found to have a similar IC₅₀ for inhibiting the viral RNA amplification (0.17 and 0.18 nM, respectively) (Fig 1E).

(2) Figure 1E

Original Figure 1E

Revised Figure 1E

(3) **Figure 2B**

Original Figure 2B

Revised Figure 2B

Comment 10:

page 5 line 15, siRNA with potent antiviral activity

Response

We appreciate the reviewer's advice. We have modified the sentence as shown below:

Original statement in P5 line 15:

These data suggest that C6G25S is a **potent siRNA** with a better safety profile beneficial for future development.

Revised statement in P6 line 23:

These data suggest that C6G25S is a siRNA **with potent antiviral activity** and a safety profile potentially beneficial for future drug development.

Comment 11:

page 6 line 2: "front", better 5 prime?

Response

We appreciate the reviewer's advice. We have modified the sentence as shown below:

Original statement in P6 line 2:

The lower part of Fig 2A shows the sequence alignment of C6 and RdRp, which is located in the **front section of** *ORF1b*.

Revised statement in P7 line 10:

The lower part of Fig 2A shows the sequence alignment of C6 and RdRp, which is located in the **5-prime region of** *ORF1b*.

Comment 12:

page 14 line 33: which SARS-CoV-2 isolate/strain/VOC

Response

We placed the virus strain information in the sub-section “Cells and viruses” of the Material and Methods section.

The virus isolates used in *in vitro* siRNA screening and IC₅₀ determination were hCoV-19/Taiwan/NTU13/2020 (A.3; EPI_ISL_422415), hCoV-19/Taiwan/NTU49/2020 (B.1.1.7; EPI_ISL_1010728), hCoV-19/Taiwan/NTU56/2021 (B.1.429; EPI_ISL_1020315), hCoV-19/Taiwan/CGMH-CGU-53/2021 (P.1; EPI_ISL_2249499), hCoV19/Taiwan/NTU92/2021 (B.1.617.2; EPI_ISL_3979387).

The viruses used in the infection of K18-hACE2 transgenic mice were hCoV-19/Taiwan/4/2020 (B; EPI_ISL_411927) and hCoV-19/Taiwan/1144/2020 (B.1.617.2; B.1.617.2; EPI_ISL_5854263).

Comment 13:

page 14: 26ff: authors must explain how exactly the animal experiments were performed in the "post exposure" setting.

Response

We appreciate the reviewer’s valuable suggestion..

Details of our post-exposure treatment are presented below:

Mice were first anesthetized and infected intranasally with SARS-CoV-2. After 30 mins of recovery, mice were placed in the chamber to undergo 30 min of inhalation treatment.

Given that the interval between siRNA delivery and virus infection was short, we have had been advised by **Reviewer 1** and replaced the “post-exposure treatment” with “co-treatment” as advised.

We have added a detailed description about the co-treatment performance in page 18 (Material and Methods section).

Original statement in Material and Methods section (page 14)

For postexposure treatment, mice were treated with aerosolized siRNA at D0 and 1-day post-infection. Two days post-infection, infected mice were sacrificed to collect lungs.

Revised statement in Material and Methods section (page 18 line 22)

For co-treatment, mice were first anesthetized with Zoletil/Dexdomitor and infected intranasally with 10^4 PFU of SARS-CoV-2. After 30 min of recovery, mice were placed in the chamber to perform 30 min of inhalation treatment (D0). Mice were treated with aerosolized siRNA at D0 and 1-day post-infection. Infected mice were sacrificed to collect their lungs at two days post-infection.

28th Jan 2022

Dear Prof. Yang,

Thank you for the submission of your revised manuscript to EMBO Molecular Medicine. I am pleased to inform you that we will be able to accept your manuscript pending the following final amendments:

1) In the main manuscript file, please do the following:

- Correct/answer the track changes suggested by our data editors by working from the attached document.
- Remove all figures and only leave figure legends.
- Add up to 5 keywords.
- Remove text highlight colour (also in the Appendix).
- Add author contributions for Tong-Young Lee and specify author contributions for Yi-Fen Chen and Yuan-Fan Chin i.e., YFeC and YFaC.
- In M&M, please specify the biosafety level for the experiments with SARS-CoV-2 by adding and amending the following sentence: All experiments with SARS-CoV-2 were performed in a ... level laboratory and with approval from...
- In M&M, provide the antibody dilutions that were used for each antibody.
- In M&M, add statistical paragraph that should reflect all information that you have filled in the Authors Checklist, especially regarding randomization, blinding, replication.
- In the reference list, citations should be listed in alphabetical order. Where there are more than 10 authors on a paper, 10 will be listed, followed by "et al.". Please check "Author Guidelines" for more information.
- <https://www.embopress.org/page/journal/17574684/authorguide#referencesformat>
- Raw data from large-scale datasets (RNA sequencing) should be deposited in one of the relevant databases and made freely available prior the publication of the manuscript. Use the following format to report the accession number of your data:

[data type]: [full name of the resource] [accession number/identifier] ([doi or URL or identifiers.org/DATABASE:ACCESSION])

Please check "Author Guidelines" for more information.

<https://www.embopress.org/page/journal/17574684/authorguide#availabilityofpublishedmaterial>

2) Conflict of interest: Rename "Conflict of interest" to "Disclosure Statement & Competing Interests". We updated our journal's competing interests policy in January 2022 and request authors to consider both actual and perceived competing interests. Please review the policy <https://www.embopress.org/competing-interests> and update your competing interests if necessary.

3) Synopsis:

- Synopsis image: Please provide a separate, high-resolution 550 px-wide x (250-400)-px high jpeg file.
- Please check your synopsis text and image and submit their final versions with your revised manuscript. Please be aware that in the proof stage minor corrections only are allowed (e.g., typos).

4) For more information: Please remove corresponding author's e-mail address. This space should be used to list relevant web links for further consultation by our readers. Could you identify some relevant ones and provide such information as well? Some examples are patient associations, relevant databases, OMIM/proteins/genes links, author's websites, etc...

5) Source data: We encourage you to include the source data for figure panels that show essential data. Numerical data should be provided as individual .xls or .csv files (including a tab describing the data). For blots or microscopy, uncropped images should be submitted (using a zip archive if multiple images need to be supplied for one panel). Please check "Author Guidelines" for more information. <https://www.embopress.org/page/journal/17574684/authorguide#sourcedata>

6) Press release: Please inform us as soon as possible and latest at the time of submission of the revised manuscript if you plan a press release for your article so that our publisher could coordinate publication accordingly.

7) Please be aware that we use a unique publishing workflow for COVID-19 papers: a non-typeset PDF of the accepted manuscript is published as "Just Accepted" on our website. With respect to a possible press release, we have the option to not post the "Just Accepted" version if you prefer to wait with the press release for the typeset version. Please let us know whether you agree to publication of a "Just accepted" version or you prefer to wait for the typeset version.

8) As part of the EMBO Publications transparent editorial process initiative (see our Editorial at <http://embomolmed.embopress.org/content/2/9/329>), EMBO Molecular Medicine will publish online a Review Process File (RPF) to accompany accepted manuscripts. This file will be published in conjunction with your paper and will include the anonymous referee reports, your point-by-point response and all pertinent correspondence relating to the manuscript. Let us know whether you agree with the publication of the RPF and as here, if you want to remove or not any figures from it prior to publication. Please note that the Authors checklist will be published at the end of the RPF.

9) Please provide a point-by-point letter INCLUDING my comments as well as the reviewer's reports and your detailed responses (as Word file).

I look forward to reading a new revised version of your manuscript as soon as possible.

Yours sincerely,

Zeljko Durdevic

***** Reviewer's comments *****

Referee #2 (Remarks for Author):

Overall, it is a very nice manuscript. I still believe that the medical relevance could be improved by improving the RNA delivery aspects. Clinical trials with inhaled naked siRNA have all failed so far. But for a scientific publication, the quality is good and my comments other than delivery are all well addressed.

The authors performed the requested editorial changes.

******* Reviewer's comments *******

Referee #2 (Remarks for Author):

Overall, it is a very nice manuscript. I still believe that the medical relevance could be improved by improving the RNA delivery aspects. Clinical trials with inhaled naked siRNA have all failed so far. But for a scientific publication, the quality is good and my comments other than delivery are all well addressed.

We are very thankful to the Reviewer 2 for the positive comments. We totally agree that if a suitable delivery system can be employed, we can further improve the efficacy and reduce medication dosage of our drug. The new generation medicines with specific delivery system are under investigation.

We are pleased to inform you that your manuscript is accepted for publication and is now being sent to our publisher to be included in the next available issue of EMBO Molecular Medicine.

Corresponding Author Name: Pan-Chyr Yang

Journal Submitted to: EMBO molecular medicine

Manuscript Number: EMM-2021-15298